# Rethinking Generative Image Pretraining:
# How Far Are We From Scaling Up Next-Pixel Prediction?

**Xinchen Yan** [1] [*]  **Chen Liang** [2] [*]  **Lijun Yu** [2]  **Adams Wei Yu** [2]  **Yifeng Lu** [2]  **Quoc V. Le** [2]

## Abstract

This paper investigates the scaling properties of autoregressive next-pixel prediction, a simple, end-to-end yet under-explored framework for unified vision models. Starting with images at resolutions of $32 \times 32$, we train a family of Transformers using IsoFLOP profiles across compute budgets up to 7e19 FLOPs and evaluate three distinct target metrics: next-pixel prediction objective, ImageNet classification accuracy, and generation-based completion where top half of the image serves as a spatial prompt. First, optimal scaling strategy is critically task-dependent. At a fixed resolution of $32 \times 32$ alone, the optimal scaling properties for image classification and image generation diverge, where generation optimal setup requires the data size grow three to five times faster than for the classification optimal setup. Second, as image resolution increases, the optimal scaling strategy indicates that the model size must grow much faster than data size. Surprisingly, by projecting our findings, we discover that the primary bottleneck is compute rather than the amount of training data. As compute continues to grow four to five times annually, we forecast the feasibility of pixel-by-pixel modeling of images within the next five years.

## 1. Introduction

Inspired by the success in language models where the data is in the form of sequence of words (Sutskever et al., 2014; Vaswani et al., 2017) or characters (Sutskever et al., 2011), a parallel line of research has explored the similar paradigm for vision. Representing each image as a sequence of pixels, prior work (Van Den Oord et al., 2016; Chen et al., 2020)

demonstrated the possibility of learning visual recognition and generation in a simple and end-to-end fashion, through next-pixel prediction. Conceptually, next-pixel prediction is highly scalable due to its *unsupervised* nature that requires no human-annotated labels. Indeed, representing each image as pixel sequence enforces the minimal architectural priors on the structure of images. However, the general paradigm of end-to-end pixel-level modeling (Van Den Oord et al., 2016; Chen et al., 2020; Ho et al., 2020; Song et al., 2021) has become less popular over the years. This is largely due to the rise of compute-efficient methods that perform multi-stage pixel generation (Ho et al., 2022; Saharia et al., 2022; Li et al., 2025) or patch-level learning with vision tokenizers (Van Den Oord et al., 2017; Razavi et al., 2019; Esser et al., 2021; Rombach et al., 2022; Chang et al., 2022; Yu et al., 2023). Despite this shift in focus, a simple question remains unanswered: "*How far are we from scaling up next-pixel prediction?*"

Admittedly, next-pixel prediction poses many challenges compared to next-token prediction in natural languages. First, pixels have very low semantic content compared to words in a sentence that usually contain rich semantic meaning. Second, pixels have complex spatial dependencies that are non-trivial to represent through sequential modeling. The color of a pixel is not only influenced by its spatial neighboring pixels, but also objects and structures in the scene that are not locally connected. Third, the complexity of next-pixel prediction grows rapidly with the image resolution. To generate a $128 \times 128$ image, an autoregressive model must predict 16,384 pixels one after another.

In this paper, we present an analysis of the scaling properties of next-pixel prediction across both image recognition and generation. We begin our study at a fixed resolution of $32 \times 32$ pixels due to its computational tractability that enables extensive experimentation. At such resolution, images start to show distinct structures and clear object interactions, and hence provides a meaningful approximation of the situation at the native image resolution. We conduct initial scaling experiments based on the *next-pixel prediction loss*. As shown in Figure 1(a), results reveal that learning on raw pixels requires substantially $(10 - 20\times)$ higher optimal token-to-parameter ratio than on text tokens. More

---

[1]Work done at Google Deepmind. [2]Google Deepmind. Correspondence to: Xinchen Yan <skywalkeryxc@gmail.com>, Chen Liang <crazydonkey@google.com>.

*Proceedings of the 43rd International Conference on Machine Learning*, Seoul, South Korea. PMLR 306, 2026. Copyright 2026 by the author(s).

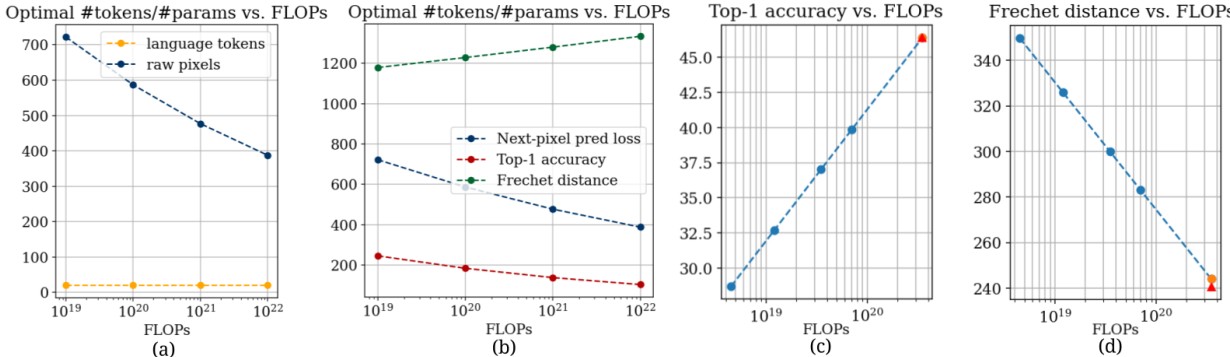

*Figure 1.* Key findings on the scaling properties of next-pixel prediction, based on training Transformers on $32 \times 32$ images. (a) Learning on raw pixels (blue line) requires $10 - 20\times$ higher optimal token-to-parameter ratio than learning on language tokens (yellow line). (b) The optimal scaling strategy varies: generation-based completion quality (Fréchet Distance, green) requires more training data optimally than classification (Top-1 accuracy, red) or the next-pixel prediction loss (blue). (c)-(d) The optimal-token and optimal-parameter setup is further verified by training models following the scaling prediction. Given 3.5e20 training FLOPs (with $5 \times$ more compute), we project to reach 46.39% accuracy (vs. 46.41% in reality) and 244 Fréchet Distance (vs. 240 Fréchet Distance in reality).

concretely, achieving a compute-optimal balance requires at least an order of magnitude more tokens per parameter compared to language models (a ratio of over $400$ vs. $20$).

This initial finding motivates a deeper investigation into three core questions. First, how do we reliably assess the performance of these models, especially at the low resolutions where extensive study is more feasible? Second, do the scaling trends derived from the simple next-pixel prediction loss align with the scaling behavior of more meaningful downstream tasks like classification and image completion? Third, how do the scaling trends shift at different resolutions? To answer these questions, we conduct a series of controlled experiments to map the scaling frontiers for three distinct metrics: next-pixel prediction loss, linear probing accuracy, and generation-based completion quality (using Fréchet Distance). At a fixed $32 \times 32$ resolution, the results reveal (see Figure 1(b)) that the optimal scaling strategy depends heavily on the target metric, with generation-based completion quality requiring a larger token-to-parameter ratio than classification or the next-pixel prediction loss. Furthermore, these scaling dynamics are not static; our study across image resolutions including $16 \times 16$ and $64 \times 64$ shows that model size must grow much faster than data size, as resolution increases.

Finally, we emphasize the **forward-looking** nature of this work. Under the data-driven *theoretical estimates*, $10^4 \times$ FLOPs increase (from $10^{20}$ to $10^{24}$) suggests that Transformers trained via next-pixel prediction could reach 80% ImageNet classification accuracy and compelling generation-based completion results. This computational requirement is currently impractical, given that well-established methods such as MAE (He et al., 2022) (with $4.6 \times 10^{20}$ FLOPs) and DINO (Caron et al., 2021) (with $2.1 \times 10^{20}$ FLOPs) achieve comparable performance with significantly less compute. On the other hand, given the

*sheer volume* of visual data available on the Internet and *unsupervised nature* of raw pixel learning, we highlight that scaling pixel-by-pixel modeling is far from data bound, but compute bound. We anticipate it will become feasible within the next five years based on the projection that the training compute for frontier AI models grows 4-5 times annually (Sevilla & Roldán, 2024).

## 2. Related Work

We situate our work within the established literature on pixel prediction models, image tokenization, and scaling laws.

**Pixel prediction models.** Autoregressive next-pixel prediction aims to model the image distribution in a tractable and scalable way. This generative learning objective has been studied on various types of neural network architectures, including RNNs (Van Den Oord et al., 2016), CNNs (Van den Oord et al., 2016; Salimans et al., 2017), and Transformers (Chen et al., 2020). Autoregressive Transformers have demonstrated remarkable capabilities at language modeling (Radford et al., 2018) through years of improvement, achieving the unification of comprehension and synthesis in a scalable manner. However, autoregressive next-pixel prediction for visual modeling falls far behind and outside mainstream approaches since the early attempt of iGPT (Chen et al., 2020). Modern Transformers are designed to handle language tokens with concise and abstract semantic meaning. Applying the same technique directly on raw pixels could suffer from the burden of long redundant sequences, especially at a high resolution. Later works of Transformers for computer vision (Dosovitskiy et al., 2020; Nguyen et al., 2025) usually operate at the level of image *patch*.

Our work also shares the same spirit with the line of research on pixel-level modeling using adversarial models (Goodfellow et al., 2014) and diffusion models (Sohl-Dickstein et al., 2015; Ho et al., 2020; Song et al., 2020; 2021). Significant

improvements in sample quality and distribution coverage were achieved by learning reverse process variances and scaling up architectures (Nichol & Dhariwal, 2021; Dhariwal & Nichol, 2021). To manage the high dimensionality of pixel space, research has branched into one-stage (Hoogeboom et al., 2023; Crowson et al., 2024) and cascaded multi-stage models (Ho et al., 2022; Saharia et al., 2022).

In the paper, we focus on the *scaling* pixel-level transformers for generative image pretraining in the form of autoregressive next-pixel prediction. Our journey starts from low resolution images at $32 \times 32$ to study scaling properties with regard to model and data sizes, while we study the impact of image resolution eventually.

**Image tokenization.** Representing each high resolution image as a sequence of pixels is a simple route which naturally fits into the modern LLM paradigm. However, the spatial redundancy and low information density per pixel has been a major obstacle for feasible autoregressive next-pixel modeling at scale. Pragmatically, it has become a popular trend to compress pixels into a more compact representations for auto-regressive modeling, known as image tokenization or *tokenization* in short. One of the key benefits is that tokenization facilitates learning and sampling for image data using Transformers, while keeping the compute budget feasible. For example, VQ-VAE (Van Den Oord et al., 2017) leverages vector quantization to map pixels into indices of a learned codebook, with convolutional encoder and decoder. VQGAN (Esser et al., 2021) introduces perceptual and adversarial losses into tokenizer training that steers the pixel compression for better perceptual quality. ViT-VQGAN (Yu et al., 2022) adopts pure Transformer architectures for patch-based tokenization. Register-based tokenization with Transformers (Yu et al., 2024; Zha et al., 2025) offer better flexibility on the token budget. Understanding models can leverage image tokens as training targets (Bao et al., 2022; Ren et al., 2024), while recent studies have started challenging the necessity of patchification or tokenization (Wang et al., 2025) in the non-generative context. Nonetheless, image tokenizers remain on the critical path for the recent autoregressive image generation models (Sun et al., 2024; Tian et al., 2024; Kondratyuk et al., 2023; Agarwal et al., 2025; Yu et al., 2023; Tang et al., 2025; He et al., 2025). In this paper, we revisit the alternative route of autoregressive next-pixel prediction and analyze its scaling potential in the next five years.

**Scaling law.** Prior works (Kaplan et al., 2020; Hoffmann et al., 2022) have established the scaling law study for language modeling, where a power-law relationship exists between the loss and the compute measured by FLOPs under the compute optimal strategy for allocating the number of parameters and the amount of training data, i.e., the best token-parameter ratio. (Henighan et al., 2020) conducted a scaling study for next-pixel prediction on images under different resolutions, but their result suffers from the same issue as (Kaplan et al., 2020) of overestimating the benefit of parameter scaling versus data scaling due to using intermediate losses of a longer training to estimate the loss of a shorter training. We follow the IsoFLOP profiles approach in (Hoffmann et al., 2022) to conduct the scaling law study, which corrected the aforementioned issue. As a result, we arrived at a slower optimal parameter scaling than theirs at $32 \times 32$ and, more interestingly, found that the optimal scaling strategy changes towards faster parameter scaling consistently when the image resolution increases. For visual understanding tasks, (Zhai et al., 2022; Dehghani et al., 2023) studied the potential of scaling ViT models to up to 22B, and (Alabdulmohsin et al., 2023; Wang et al., 2025) extended the study of optimal scaling strategy to other hyperparameters beyond token-parameter ratio such as model width and depth. In contrast, we attempted to study and compare the scaling properties of both recognition and generation tasks. We focus on the optimal data and model sizes since it is the most important hyperparameter for scaling and leave the study of other hyperparameters as future work.

While prior work has established scaling laws for language and explored autoregressive models for vision, a rigorous analysis of how *raw* next-pixel prediction scales across recognition and generation tasks remains under-explored. In the following section, we outline our methodology to directly address this gap.

## 3. Method

### 3.1. Learning next-pixel prediction

Given an image $x \in [1, K]^{s \times s}$, we model the probability $\Pr(x)$ by autoregressively predicting the next-pixel $\Pr(x) = \prod_{i=1}^{s \times s} \Pr(x_i | x_1, x_2, \cdots, x_{i-1})$. Here, $x_1, x_2, \cdots, x_{s \times s}$ represents the sequence of pixels on the image following the raster order. Additionally, we assume each term takes the form of a simple categorical distribution of size $K$. The next-pixel prediction model is trained using the standard negative log-likelihood objective on a large-scale image dataset. $\mathcal{L}_{\text{train}} = \mathbb{E}_{x \in X}[-\log \Pr(x)]$.

### 3.2. Transformer model architecture

We use a standard family of modern Transformer architectures (Anil et al., 2023; Dubey et al., 2024; Team et al., 2024), each equipped with pre-normalization using RMSNorm (Zhang & Sennrich, 2019), one-dimensional RoPE (Su et al., 2024), and Gated Linear Units using GELU activations (Shazeer, 2020) in the feed-forward module.

$$\hat{Z}^l = \text{MHSA}(\text{RMSNorm}(Z^{l-1})) + Z^{l-1} \qquad (1)$$

$$Z^l = \text{FFN}_{\text{GEGLU}}(\text{RMSNorm}(\hat{Z}^l)) + \hat{Z}^l \qquad (2)$$

Specifically, we conduct studies on model architectures at five different scales, ranging from 10-million parameters

| Model | Layers (N) | Hidden dim ($d$) | MLP dim |
|---|---|---|---|
| $S^{--}$: 10M | 12 | 256 | 768 |
| $S^-$: 28M | 16 | 384 | 1024 |
| S: 77M | 24 | 512 | 1408 |
| $B^-$: 227M | 32 | 768 | 2048 |
| B: 449M | 36 | 1024 | 2688 |

*Table 1.* Specification of five model architectures at different scales. Except for the smallest Transformer $S^{--}$ models with 4 attention heads, the rest of the Transformers have 8 attention heads. We conduct our preliminary scaling studies with four scales ($S^-$, S, $B^-$, B) on $32 \times 32$ and multi-resolution scaling studies with all five scales. We use Gated Linear Units (Shazeer, 2020) and adapt the MLP dimensions to match the model size with iGPT-S and iGPT-M (Chen et al., 2020).

to 449-million parameters. We use the Transformer implementation in the Simply codebase (Liang et al., 2025) for our study. Details of the Transformer architectures have been included in Table 1. Note that our 77-million and 449-million parameters model are designed to be very comparable to iGPT-S and iGPT-M models (Chen et al., 2020), respectively. The main difference is the designs including RoPE and GEGLU to align with modern Transformer architectures.

### 3.3. IsoFLOP profiles

By constructing IsoFLOP profiles (Hoffmann et al., 2022) given a target FLOPs budget, we can estimate the optimal tokens and optimal model size. To achieve the goal, we first vary the model size for a fixed set of five different training FLOPs budget. For each base Transformer model architecture described in Table 1, we construct six additional IsoFLOP variants by perturbing model depth and width, while keeping the budgeted training FLOPs the same across IsoFLOP variants (see Table 2 for details). This results in $35 (= 5 \times 7)$ different training runs per resolution (105 runs total). Although smaller in absolute scale than language modeling studies (Hoffmann et al., 2022), this concentrated run density ensures identification of compute-optimal minima within our tested FLOPs regime.

| Variant | Layers (N) | Hidden dim (d) | MLP dim |
|---|---|---|---|
| Var-0 | 2 $\times$ | | |
| Var-1 | | 2 $\times$ | 2 $\times$ |
| Var-2 | 2 $\times$ | 2 $\times$ | 2 $\times$ |
| Var-3 | 0.5 $\times$ | | |
| Var-4 | | 0.5 $\times$ | 0.5 $\times$ |
| Var-5 | 0.5 $\times$ | 0.5 $\times$ | 0.5 $\times$ |

*Table 2.* IsoFLOPs variants: for each Transformer architecture in Table, we construct six IsoFLOPs variants by perturbing the depth and width of the base Transformer, while maintaining the same budgeted FLOPs across variants. Blank entries in the table indicate that those hyperparameters are kept the same as the baseline model.

Following (Hoffmann et al., 2022), we fit a parabola to each IsoFLOPs curve to estimate the model size that leads to the minimum loss, and then a power law between FLOPs and compute-optimal Transformer model size $N_{\text{opt}} \propto C^a$ and number of training tokens $D_{\text{opt}} \propto C^b$, respectively. See figure 2 for the results.

**Different tasks.** To further answer the question "do the scaling trends derived from the simple next-pixel prediction loss align with the scaling behavior of downstream tasks such as classification and image completion?", we repeat the study with a different measure including linear probing accuracy for classification and image completion-based Fréchet Distance.

**Different resolutions.** In contrast to scaling large language models (LLMs), the sequence length in the next-pixel prediction setup grows quadratically with the image resolutions. To quantify the scaling trend across different image resolutions, we consider the optimal number of images (as $D_{\text{opt}}^{\text{pp}} := \frac{1}{s^2} D_{\text{opt}}$) by dividing the number of pixels within each corresponding image resolution. Similarly, we consider the optimal number of parameters per pixels (as $N_{\text{opt}}^{\text{pp}} := \frac{1}{s^2} N_{\text{opt}}$) by dividing the model size with the number of pixels per image resolution. Compared to the classic ones, these new measures ($N_{\text{opt}}^{\text{pp}}$ and $D_{\text{opt}}^{\text{pp}}$) do not impact the slope of linear fitting but only the bias.

### 3.4. Training

We perform pre-training on JFT-300M (Sun et al., 2017; Chollet, 2017; Hinton et al., 2015). Compared to the ImageNet ILSVRC 2012 training set, JFT-300M dataset features a heavily long-tailed distribution. This naturally skewed distribution much more accurately reflects open-world visual frequencies, encompassing a vast variety of complex scenes, natural landscapes, text-heavy images, and diverse objects. We utilized this specific mixture precisely to ensure our scaling laws generalized to *in-the-wild* visual modeling, rather than overfitting to an ImageNet training set. One pass over the dataset is equivalent to going through over $300B$ pixels, given a resolution of $32 \times 32$ pixels. We apply standard Inception-style random cropping (Szegedy et al., 2015) and flip the image horizontally (with 50% chance) over the course of training.

We train all our models with a batch size of 512 using Adam optimizer with $\beta_1 = 0.9$ and $\beta_2 = 0.95$. We warm-up the learning rate from 0 to the maximum learning rate 0.001 over the first 1,000 steps, and then decay it $10\times$ using a cosine schedule. We implement our experiment using the Simply codebase (Liang et al., 2025).

### 3.5. Downstream Evaluations

We conduct two types of downstream evaluations, namely, image classification and image completion. Our preliminary

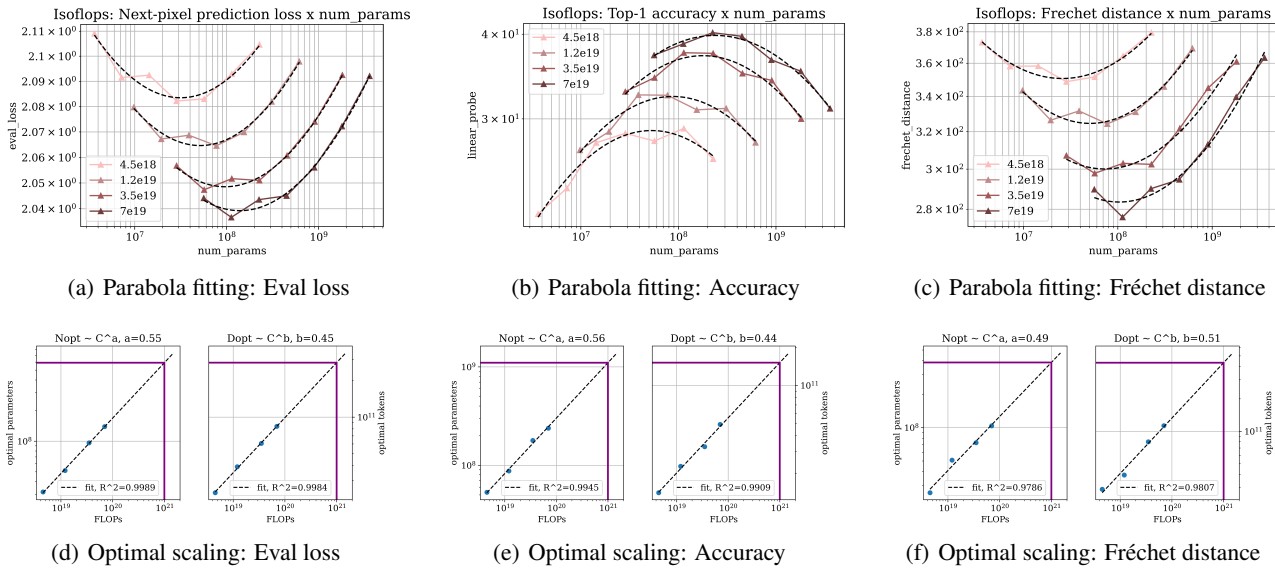

(a) Parabola fitting: Eval loss      (b) Parabola fitting: Accuracy      (c) Parabola fitting: Fréchet distance

(d) Optimal scaling: Eval loss      (e) Optimal scaling: Accuracy      (f) Optimal scaling: Fréchet distance

*Figure 2.* Scaling properties prediction given image resolutions at $32 \times 32$: next-pixel prediction loss (See subfigure (a) and (d)), ImageNet classification accuracy (See subfigure (b) and (e)), and image completion-based Fréchet Distance (See subfigure (c) and (f)). We report the best-layer linear probing accuracy. We estimate Fréchet Distance between 2,048 reference images at $32 \times 32$ and corresponding 8,192 generated images.

analysis is conducted at $32 \times 32$ resolution. We further investigate the impact of resolution on scaling dynamics by studying $16 \times 16$ and $64 \times 64$ resolution.

**ImageNet classification.** For each pre-trained Transformer model, we perform linear probing using activations from different Transformer layers and report the best-layer linear probing accuracy (Chen et al., 2020; Caron et al., 2021) among them. Specifically, we freeze the Transformer backbone and extract activations from each layer given the input pixel sequence. Given the frozen features as input, we train a linear head as a simple 1000-way classifier using Stochastic Gradient Descent (SGD) for 600 epochs with a batch size of 4,096 images. We perform grid search over different learning rates and report the performance as the best linear probing accuracy on a combination of learning rate and the Transformer depth where we extracted feature activations from. Such exhaustive search is critical to obtain a robust metric in our IsoFLOP study, as the optimal linear probing learning rates vary with different Transformers.

**Image generation-based completion.** We report the Fréchet Distance as our metric for image generation-based completion quality where the top half of the image serves as a spatial prompt. Specifically, we collect images from ImageNet validation set and mask out the bottom half image. We use the unmasked top image as initialization and let the pre-trained Transformer auto-regressively predict the bottom half image one pixel at a time (Van Den Oord et al., 2016; Chen et al., 2020). In our single resolution experiment (at $32 \times 32$), we generate four images for each of

the real image and measure the Fréchet Distance between 2,048 reference images and 8,192 generated images at the same $32 \times 32$ resolutions to confirm the predictability of the power-law prediction (see Figure 1(c) and Figure 1(d)). To facilitate comparisons across different resolutions, we keep the ground-truth reference images at the native resolutions, and report the Fréchet Distance between 10,000 reference images and 10,000 generated images, which leads to stable estimate. We use DINOv2 backbone (Oquab et al., 2024) to extract features for Fréchet Distance, as it is reportedly more aligned with human preference (Stein et al., 2023).

## 4. Experiments

### 4.1. Do pixels play by the same rules as text?

Yes, the raw pixel prediction follows predictable scaling trends just like text, but it is far less efficient. As each individual pixel alone carry very minimal semantic information compared to text, we estimate that models require 10-20× more tokens per parameter to learn effectively from raw pixels than from language tokens.

We conduct preliminary studies through scaling the next-pixel prediction loss. As illustrated in Figure 2(a) and Figure 2(d), we can fit a power law relationship between training FLOPs and the optimal model size $N_{\text{opt}}$ (number of parameters) and data size $D_{\text{opt}}$ (number of training tokens / pixels). Specifically, the eval loss optimal Transformer model and data size is given by $N_{\text{opt}} \propto C^{0.55}$ and $D_{\text{opt}} \propto C^{0.45}$, respectively. On raw pixels alone at $32 \times 32$ resolution , the model size growth will have a more direct impact to the next-pixel prediction loss than the training dataset size

growth. We further compute the optimal token-to-parameter ratio with respect to the training FLOPs and compare with typical language models (Hoffmann et al., 2022). As illustrated in Figure 1(a), the results reveal that learning on raw pixels require substantially $(10 - 20\times)$ more data than on text tokens even beyond FLOPs budget of $10^{21}$. This suggests that the semantic level of a single pixel, even in a low-resolution $32 \times 32$ image, is substantially lower than that of a language token, which represents a more compact and meaningful unit of information. Intuitively, a single word like *cat* is a highly compressed symbol, carrying a massive amount of abstract information: it is an animal, it has fur, it meows, it has whiskers. A single pixel contains little semantic information on its own, since signal value of a pixel could be part of a cat, a car, or looking at the sky.

In summary, our results have demonstrated that the optimal scaling trend of the next-pixel prediction is indeed *predictable* using the framework (Kaplan et al., 2020; Hoffmann et al., 2022) that has been well established in language models.

### 4.2. Optimal scaling transfer to downstream tasks?

Not in a simple way. At the fixed resolution of $32 \times 32$ alone, the optimal scaling strategy established by the pixel prediction loss is sub-optimal for image generation. Specifically, achieving good image generation-based completion quality requires a much more data-heavy approach where the dataset size grows faster than the model size.

**Image recognition.** We follow the process described in Sec. 3.5 where we report the best-layer linear probing accuracy as the robust measure for scaling study. As illustrated in Figure 2(b) and Figure 2(e), the power law between training FLOPS and classification optimal Transformer model size and number of training tokens are given by $N_{\text{opt}} \propto C^{0.56}$ and $D_{\text{opt}} \propto C^{0.44}$, respectively.

**Image generation-based completion.** Beyond the image recognition, we further extend our study to structured and dense understanding task of image generation-based completion. It is worth noting that the paradigm of the next-pixel prediction naturally accommodates left-to-right image generation (Van Den Oord et al., 2016; Chen et al., 2020), where the top half of the image serves as a spatial prompt for generation. Therefore, we use the *image completion* as a proxy task to measure the image generation quality.

As illustrated in Figure 2(c)(f), the power law between training FLOPs and generation optimal Transformer size and number of training tokens are given by $N_{\text{opt}} \propto C^{0.49}$ and $D_{\text{opt}} \propto C^{0.51}$, respectively.

**Is power-law prediction precise?** We specifically trained models with $5\times$ more compute by following the optimal-token and optimal-parameter prediction. Given scaling prediction for FLOPs budget of $3.5 \times 10^{20}$ from the classification optimal setup, we project to reach 46.39% top-1 accuracy on the ImageNet benchmark. We therefore train a model following the scaling prediction. We have verified that the scaling prediction is very precise, given the model achieves 46.41% accuracy. Similarly, given scaling prediction for FLOPs budget of $3.5 \times 10^{20}$ from generation optimal setup, we project to reach 244 Fréchet Distance. We further verify the prediction is precise by training a model, which achieves 240 Fréchet Distance. See Figure 1(c) and Figure 1(d) for more details.

**Scaling optimalities vary across tasks.** As illustrated in Figure 1(b), we compare optimal token-to-parameter ratios estimated by independent IsoFLOP profiles across next-pixel prediction loss, top-1 accuracy on ImageNet classification benchmark, and image completion-based Fréchet Distance. It is worth noting that the estimated optimal scaling trends vary across tasks.

Under image resolution of $32 \times 32$, the optimal scaling strategy that is effective for next-pixel prediction loss roughly aligns with the optimized strategy for improving image classification accuracy to some extent (evidenced by similar growth rate of $C^a$ and $C^b$). The performance of image classification models tends to plateau, reaching a point where a sufficient amount of data has been provided and further data increases offer little benefit. It becomes more effective to invest in model scaling to learn the abstract concepts. On the other hand, a scaling strategy that is effective for next-pixel prediction loss is suboptimal for improving generation quality. For example, at a fixed resolution of $32 \times 32$ pixels, the training data growth becomes more effective than model size growth for the image generation task compared to the image classification task. Intuitively, optimizing the image generation metric requires training on more diverse data points to capture variations ranging from abstraction level (e.g., which class it is) to low-level cues such as textures. It stands to reason that structured, dense prediction tasks such as image generation demand vast and diverse training data. This is necessary for a model to learn the higher-order dependencies across pixels and to account for rare, long-tail visual distributions.

### 4.3. Scaling optimalties shift with image resolutions?

Yes, the optimal scaling strategy changes as image resolution increases. The scaling trend shifts from balancing model and data size at $32 \times 32$ resolutions to heavily favoring larger models over more data at higher resolutions.

Image resolution is a fundamental property of visual data that has no direct equivalence in natural language. So far, the majority of our experiments were conducted at a fixed

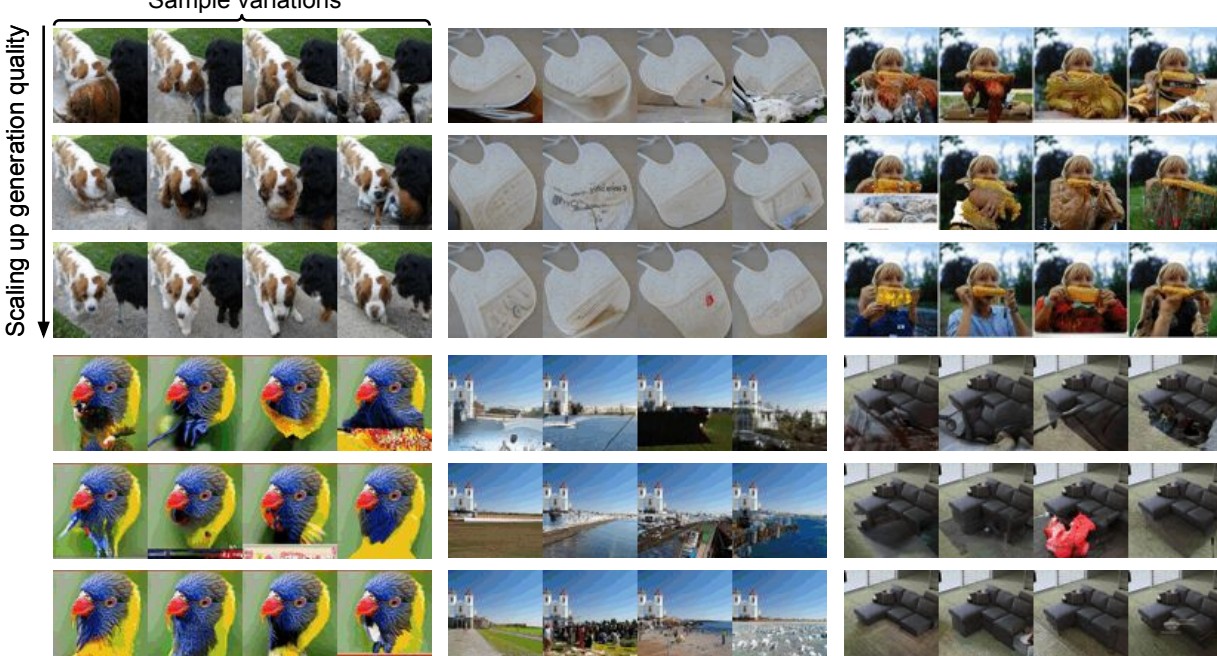

*Figure 3.* Scaling image generation-based completion at $64 \times 64$. In each visualization group, every row corresponds to a different model architecture scale and every column represents a different sample from the model. In each image example, the unmasked top half image is provided as initialization and we autoregressively predict the bottom half image one pixel at a time. Zoom in for a better view.

$32 \times 32$ resolution, which was a necessary choice to ensure computational tractability across our many experimental runs. However, this controlled setting assumes a lossy compression in the raw pixel representation. Specifically, we address this by conducting ablation studies on downstream task performances across different image resolutions. In this multi-resolution experiment, we include all five model architecture scales in Table 1 with their corresponding IsoFLOP variants and train each model at different image resolutions: $16 \times 16$, $32 \times 32$, and $64 \times 64$. It is crucial to note that while the Transformer architectures themselves remained the same (e.g., S: 77M always had the same parameter count), the input sequence length grows quadratically with the resolution change. We then repeat our evaluation process, generating 10,000 samples for measuring Fréchet Distance and extracting Transformer layer activations to estimate the best-layer top-1 accuracy at these new resolutions. This comparison is meaningful as downstream tasks performance serves as a comparable measure across different resolutions.

**Image classification vs. generation-based completion.** As consistently illustrated in Figure 4(a) and Figure 4(c), training models on higher resolutions contribute positively to downstream performance. For image classification, we see noticeable gains by switching from $16 \times 16$ to $32 \times 32$ but only slight improvements going from $32 \times 32$ to $64 \times 64$ with $> 1e20$ FLOPs budget. This suggests that the image resolution increase contributes less beyond a certain point at $32 \times 32$ for the image classification task (ImageNet bench-

mark). For image generation-based completion, the scaling trends have not been saturated around $32 \times 32$, as clear improvements have been observed by increasing the resolution from $32 \times 32$ to $64 \times 64$. Intuitively, as image resolution increases, the information density per pixel reduces accordingly but with more complex and authentic visual structure across individual pixels to model. Abstraction and semantic content can be captured effectively at lower resolutions, while fine-grained textures require higher resolutions. The results demonstrated that extra complexity and information in the $64 \times 64$ does not bring about significant gains in ImageNet classification. On the other hand, higher image resolutions can still lead to significant reductions in Fréchet Distance through model and data scaling.

**Model vs. data scaling?** We plot the scaling trends on the classification optimal and generation optimal setup in Figure 4(b) and Figure 4(d), respectively. Model scaling becomes more effective at higher resolutions, while data scaling becomes less effective at higher resolutions. We quantify this by discounting the input sequence length per image. First, we discover that the optimal model size per-pixel (as $N_{\text{opt}}^{\text{pp}}$) grows more rapidly (evidenced by steady increase of fitted exponent a) with the compute budget as the image resolution increases. This suggests that investing compute into larger models is an effective and important strategy for handling the added complexity of higher-resolution images. Second, the scaling trend (specifically, the slope of the line as compute increases) shows that the optimal data size per-

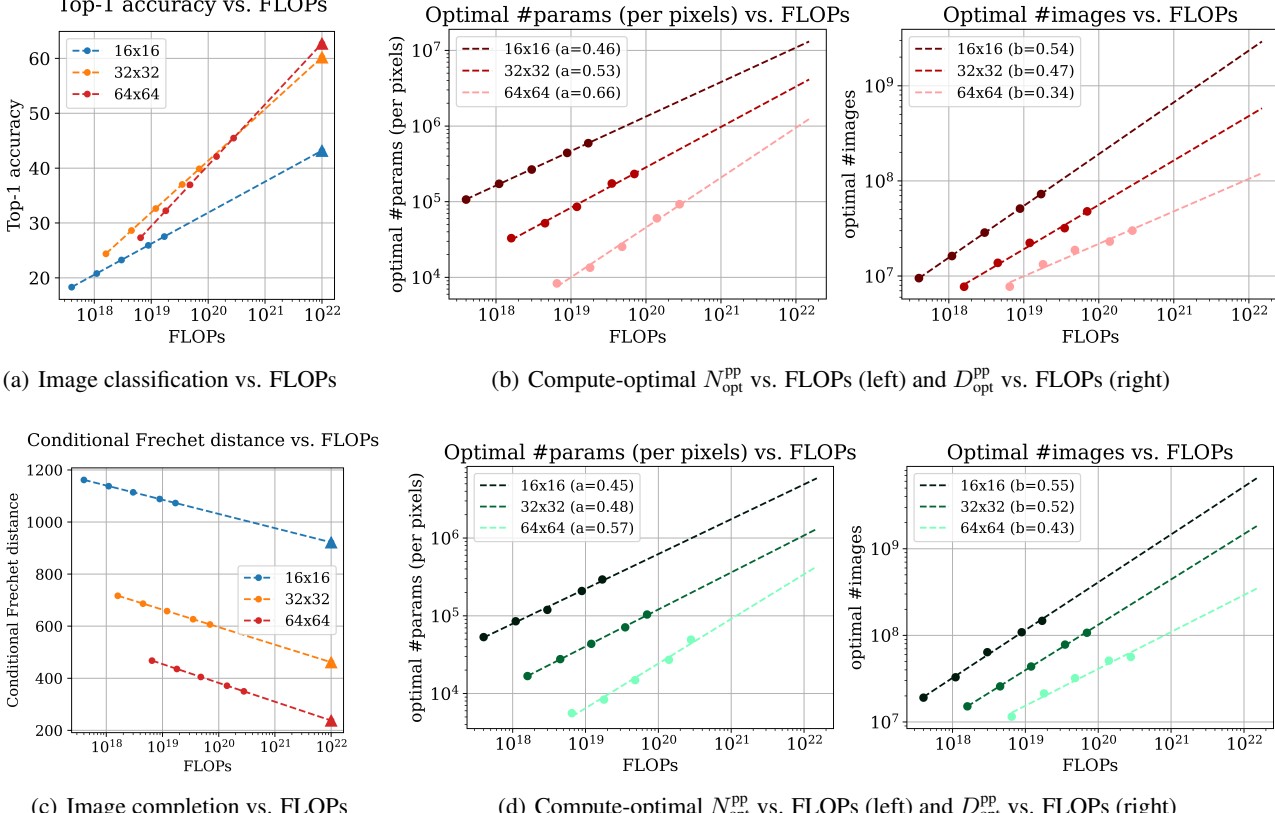

*Figure 4.* Optimal model / data scaling predictions vs. FLOPs across different image resolutions. We keep the ground-truth reference images at the native resolutions, and report the Fréchet Distance between 10,000 reference images and 10,000 generated images. Specifically, we use $10,000$ samples for cross-resolution comparisons to ensure a stable estimate for Fréchet Distance. IsoFLOP profiles: (i) From left-to-right, $16 \times 16$ IsoFLOPs are given by $4 \times 10^{17}$, $1.1 \times 10^{18}$, $3 \times 10^{18}$, $8.9 \times 10^{18}$, $1.7 \times 10^{19}$; (ii) From left-to-right, $32 \times 32$ IsoFLOPs are given by $1.6 \times 10^{18}$, $4.5 \times 10^{18}$, $1.2 \times 10^{19}$, $3.5 \times 10^{19}$, $7 \times 10^{19}$; (iii) From left-to-right, $64 \times 64$ IsoFLOPs are given by $6.5 \times 10^{18}$, $1.8 \times 10^{19}$, $4.8 \times 10^{19}$, $1.4 \times 10^{20}$, $2.8 \times 10^{20}$.

pixel (or number of images $D_{\text{opt}}^{\text{pp}}$) grows noticeably slower at higher resolutions. Intuitively, learning autoregressive next-pixel modeling is inherently a multi-task process. Training at a higher resolution naturally includes the information from lower-resolution sub-sequences. Consequently, as the amount of training data becomes adequate, the optimal scaling shifts more towards model size, because the need for more data grows relatively slowly at higher resolutions.

### 4.4. How far are we from raw next-pixel prediction?

While currently impractical due to massive computational costs, raw pixel-by-pixel modeling is a viable path to achieving competitive performance within the next five years. The primary bottleneck is compute rather than the availability of training data.

We predict the required training FLOPs to reach competitive ImageNet classification accuracy and image generation-based completion metric. As presented in Table 3, under a theoretical estimate by extrapolating from the higher end of IsoFLOP variants (or $10^{20}$), one can achieve beyond $80\%$

classification accuracy with a simple next-pixel prediction objective. At the compute budget of $10^{24}$ FLOPs, the optimal data projection (or half billion images) at $64 \times 64$ roughly matches the dataset size ( 300M images) in use. The optimal model size projection (80.72B model) is on the higher end for vision models at the moment given that well-established methods such as MAE (He et al., 2022) and DINO (Caron et al., 2021) achieve comparable performance with significantly smaller model. Similarly, as shown in Table 4, under a theoretical estimate by extrapolating from the higher end of our IsoFLOP variants (or $10^{20}$), one can significantly reduce completion-based Fréchet Distance measure below 100 by training models with next-pixel prediction objective. As a reference, the Fréchet Distance between ground-truth at native resolution and images downscaled to $64 \times 64$ resolution is measured 49.6. Given $10^{24}$ FLOPs, the optimal data projection (or 2.06B images) at $64 \times 64$ requires 7 times growth of the dataset we used, while the projection (or 16.4B images) at $32 \times 32$ requires 54 times growth. Given the sheer volume of visual data available on the Internet, we believe scaling pixel-by-pixel modeling is

| Acc ↑ ($N_{opt}$, $D_{opt}^{pp}$) ‖ FLOPs | $10^{22}$ | $10^{23}$ | $10^{24}$ |
|---|---|---|---|
| $16 \times 16$ Projection | 43.1% (2.78B, 2.33B) | 48.8% (7.95B, 8.18B) | 54.4% (22.73B, 28.6B) |
| $32 \times 32$ Projection | 60.1% (3.39B, 0.58B) | 69.6% (11.6B, 1.4B) | 79.0% (39.64B, 4.1B) |
| $64 \times 64$ Projection | 62.7% (3.88B, 0.1B) | 73.8% (17.7B, 0.22B) | 84.9% (80.72B, 0.5B) |

*Table 3.* Predicted ImageNet classification by growing budgeted training FLOPs under data-driven *theoretical estimates*. The corresponding **model size** ($N_{opt}$) and **number of images** ($D_{opt}^{pp}$) under the estimates are provided.

| FD ↓ ($N_{opt}$, $D_{opt}^{pp}$) ‖ FLOPs | $10^{22}$ | $10^{23}$ | $10^{24}$ |
|---|---|---|---|
| $16 \times 16$ Projection | 921.7 (1.25B, 5.19B) | 867.1 (3.51B, 18.51B) | 812.6 (9.86B, 65.97B) |
| $32 \times 32$ Projection | 461.3 (1.1B, 1.47B) | 394.0 (3.3B, 4.92B) | 326.7 (9.90B, 16.4B) |
| $64 \times 64$ Projection | 237.4 (1.39B, 0.29B) | 165.2 (5.24B, 0.77B) | 92.9 (19.67B, 2.06B) |

*Table 4.* Predicted completion-based Frechet Distance on ImageNet validation subset by growing budgeted training FLOPs under data-driven *theoretical estimates*. The corresponding **model size** ($N_{opt}$) and **number of images** ($D_{opt}^{pp}$) under the estimates are provided.

not bottlenecked by the data but the compute. Furthermore, with our scaling trend projection and the estimated compute growth 4-5 times annually, we anticipate learning on raw pixels will become a feasible direction in the next five years.

## 5. Limitations

While this study establishes a systematic scaling baseline for autoregressive next-pixel prediction, several technical and conceptual constraints define the boundaries of the current empirical findings.

**Scaling extrapolation and empirical boundaries.** While scaling trajectories were empirically verified up to $3.5 \times 10^{20}$ FLOPs, the projection to $10^{24}$ FLOPs represents a four-order-of-magnitude ($10,000 \times$) jump beyond the tested regime. These theoretical estimates assume the continued stability of power-law exponents; however, scaling beyond towards multi-billion parameter models may introduce qualitative shifts in scaling dynamics or emergent structural bottlenecks. Furthermore, empirical evidence is strictly bounded by resolutions ranging from $16 \times 16$ to $64 \times 64$ pixels. Given that sequence lengths in pixel-level modeling grow quadratically with image resolution, modeling high-definition visual data involves computational complexities and context lengths that may fundamentally alter the optimization landscape.

**Downstream evaluation metrics.** The scope of current downstream evaluations is restricted to ImageNet classification and generation-based completion. While these benchmarks demonstrate both discriminative and generative capabilities, they do not yet incorporate more complex visual tasks such as object detection or semantic segmentation, which are characteristic of fully unified vision models. Benchmarking these tasks was not computationally practical within the evaluated resolution regime, as they typically require higher-resolution inputs, often exceeding $224 \times 224$, to be meaningful. Additionally, the use of completion-based

Fréchet Distance (FD) for measuring generation quality differs from the unconditional FID metrics prevalent in latent-space generative research. While this protocol provides a stable semantic anchor for assessing generative capacity on held-out validation sets, it limits direct performance comparisons with latent-space diffusion models.

## 6. Conclusion and Future work

This work addresses a fundamental question towards raw pixels modeling: what is the true cost of directly modeling raw pixels? We demonstrate that autoregressive next-pixel prediction is a viable path towards end-to-end learning from raw pixels. Indeed, it follows predictable scaling trends just like text. First, optimal scaling strategy is task-dependent. Optimizing for image generation demands the data size must grow 3-5 times faster than that for image classification. This highlights a challenge towards unifying image recognition and generation from the compute-optimal perspective. Second, these scaling trends are image resolution-dependent. As image resolution increases, the optimal strategy demands that model size must grow much faster than data size. This suggests that high-resolution images inherently provide a richer and more complex learning signal, reducing the marginal value of new data points.

Possible future work directions include further investigating scaling dynamics on images beyond $64 \times 64$ resolutions, innovations to improve training efficiency, different training objectives (e.g., diffusion), and discovering methods to unify the scaling trends across various downstream tasks.

## Acknowledgements

We thank Da Huang and Andrew Li for their helpful discussions about the implementations, and Weizhe Hua for discussions of the preliminary results and ideas. Special thanks to Lisa Patel, Minmin Chen, and Ruben Villegas for providing constructive feedback.

## Impact Statement

This paper presents work whose goal is to advance the field of machine learning. There are many potential societal consequences of our work, none of which we feel must be specifically highlighted here.

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

# A. Implementation details

## A.1. Transformer blocks

This section provides the JAX-based code structure for the Transformer architectures used in the study.

The code defines a `TransformerBlock` class that implements a standard modern Transformer architecture. It utilizes `LayerNorm` for pre-normalization (RMSNorm), followed by `Attention` and `FeedForward` modules with residual connections. `LayerNorm` module is implemented as a root mean square layer normalization (RMSNorm) to improve training stability. `FeedForward` module uses Gated Linear Units (GLU) with GELU activations, matching the architecture of modern models like PaLM 2 and Llama 3. `Attention` module is equipped with one-dimensional Rotary Positional Embeddings (RoPE) and support for multi-head self-attention.

```
1  class TransformerBlock:
2    """Basic Transformer block module."""
3    model_dim: int
4    n_heads: int
5    per_head_dim: int
6    ffn_expand_dim: int
7
8    def setup(self):
9      self.pre_ln_0 = LayerNorm(dim=self.model_dim)
10     self.pre_ln_1 = LayerNorm(dim=self.model_dim)
11     self.attn = Attention(self.model_dim, self.n_heads, self.per_head_dim)
12     self.ffn = FeedForward(model_dim=self.model_dim, ffn_expand_dim=self.ffn_expand_dim)
13
14   def apply(self, params, segmend_ids, segment_positions):
15     x_res = x
16     x = self.pre_ln_0.apply(params['pre_ln_0'], x)
17     x = self.attn.apply(params['attn'], x, segment_ids, segment_positions)
18     x += x_res
19
20     x_res = x
21     x = self.pre_ln_1.apply(params['pre_ln_1'], x)
22     x = self.ffn.apply(params['ffn'], x)
23     x += x_res
24     return x
25
26
27 class LayerNorm:
28   """LayerNorm module."""
29   dim: int
30   axis: int = -1
31   epsilon: float = 1e-6
32
33   def apply(self, params, x):
34     x -= jnp.mean(x, axis=self.axis, keepdims=True)
35     var = jnp.mean(jnp.square(x), axis=self.axis, keepdims=True)
36     x *= jax.lax.rsqrt(var + self.epsilon)
37     x *= params['scale'] + jnp.array(1.0)
38     x += params['bias']
39
40
41 class FeedForward:
42   """MLP module."""
43   model_dim: int
44   ffn_expand_dim: int
45
46   def setup(self):
47     weight_shape=[self.model_dim, self.ffn_expand_dim]
48     self.ffn_0 = EinsumLinear(
49         eqn='io,...i->...o', weight_shape=weight_shape, bias_term='o')
50     self.ffn_0_gate = EinsumLinear(
51         eqn='io,...i->...o', weight_shape=weight_shape, bias_term='o')
```

```python
52    self.ffn_1 = EinsumLinear(
53        eqn='io,...i->...o', weight_shape=weight_shape[::-1], bias_term='o')
54
55  def apply(self, params, x):
56    projected_x = self.ffn_0.apply(params['ffn_0'], x)
57    gate = self.ffn_0_gate.apply(params['ffn_0_gate'], x)
58    x = self.ffn_1.apply(params['ffn_1'], jax.nn.gelu(gate) * projected_x)
59    return x
60
61
62 class Attention:
63  """Basic multi-head self-attention module."""
64  model_dim: int
65  n_heads: int
66  per_head_dim: int
67
68  def setup(self):
69    self.per_dim_scale = PerDimScale(self.per_head_dim)
70    q_shape = [self.model_dim, self.n_heads, self.per_head_dim]
71    kv_shape = q_shape
72    self.q_proj = EinsumLinear(eqn='ihd,...i->...hd', weight_shape=q_shape)
73    self.k_proj = EinsumLinear(eqn='ihd,...i->...hd', weight_shape=kv_shape)
74    self.v_proj = EinsumLinear(eqn='ihd,...i->...hd', weight_shape=kv_shape)
75    self.o_proj = EinsumLinear(eqn='ihd,...hd->...i', weight_shape=q_shape)
76
77  def apply(self, params, x, segment_ids, segment_positions):
78    # q: [batch_size, seq_len, n_heads, per_head_dim]
79    q = self.q_proj.apply(params['q_proj'], x)
80    # k: [batch_size, seq_len, n_heads, per_head_dim]
81    k = self.k_proj.apply(params['k_proj'], x)
82    # v: [batch_size, seq_len, n_heads, per_head_dim]
83    v = self.v_proj.apply(params['v_proj'], x)
84
85    q = rotary_positional_embedding(q, segment_positions=segment_positions)
86    k = rotary_positional_embedding(k, segment_positions=segment_positions)
87
88    q = self.per_dim_scale.apply(params['per_dim_scale'], q)
89    q = q / jnp.sqrt(self.per_head_dim)
90
91    q = einops.rearrange(q, '... (n_kv_heads g) h -> g n_kv_heads h', n_kv_heads=self.
     n_kv_heads)
92
93    q_seq_len = q.shape[1]
94    kv_seq_len = k.shape[1]
95
96    mask = create_mask(
97      segment_positions=segment_positions,
98      kv_segment_positions=kv_segment_positions,
99      segmend_ids=segment_ids,
100      kv_segment_ids=kv_segment_ids,
101      window_size=0)
102    # Add the group and head dimension.
103    mask = einops.rearrange(mask, 'b l1 l2 -> b 1 1 l1 l2')
104
105    # q: [batch_size, seq_len, n_groups, self.n_kv_heads, self.per_head_dim]
106    # k, v: [batch_size, seq_len, self.n_kv_heads, self.per_head_dim]
107    output, attn_mat = attn(
108      q, k, v, mask, attn_soft_cap=50.0)
109    output = einops.rearrange(
110      output, '... n_groups n_kv_heads i -> ... (n_kv_heads n_groups) i')
111    output = self.o_proj.apply(params['o_proj'], output)
112    return output
```

# B. Additional results

This section presents detailed scaling curves and qualitative data for various resolutions and model depths.

## B.1. IsoFLOP curves for multi-resolution experiments

These figures illustrate the IsoFLOP profiles used to determine optimal scaling for image classification and generation across different resolutions. Figure 5 shows IsoFLOP for Top-1 accuracy (top row) and Fréchet Distance (bottom row). Fittings indicate optimal parameters ($a = 0.46$ for accuracy; $a = 0.45$ for FD) and optimal tokens ($b = 0.54$ for accuracy; $b = 0.55$ for FD). Figure 6 displays similar scaling laws at $32 \times 32$. Compared to $16 \times 16$, the optimal model size grows faster with compute budget ($a = 0.53$ for accuracy; $a = 0.48$ for FD). As illustrated in Figure 7, at this higher resolution, the shift toward model scaling is most pronounced, with $a = 0.66$ for accuracy and $a = 0.57$ for Fréchet Distance.

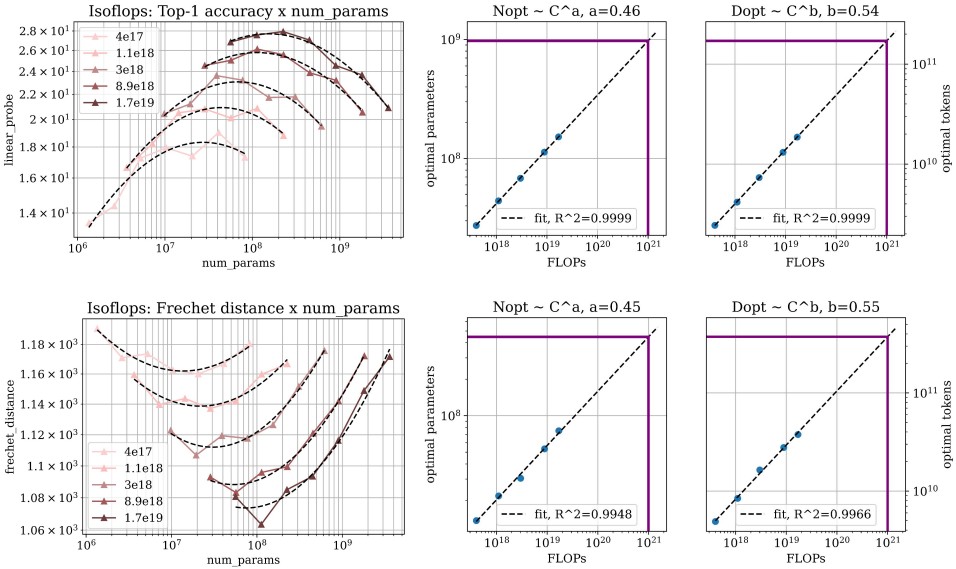

*Figure 5.* 16x16 IsoFLOPs, Top-1 accuracy (top row), Frechet Distance (bottom row) between generated samples and ground-truth images at native resolution.

## B.2. Per-layer probing diagram

These bar charts analyze the linear probing accuracy across different layers of the Transformer for various compute budgets and model sizes. Figure 8 evaluates models with 8, 16, and 32 layers. Best performance is generally found in middle-to-late layers. Figure 9 shows results for models up to 48 layers, with a peak accuracy of 32.73%. Figure 10 continues the trend for models up to 64 layers, peaking at 37.55% accuracy. Figure 11 The highest compute budget shown in the per-layer analysis, reaching 39.75% accuracy at Layer 16 of a 36-layer model.

## B.3. Qualitative generation results

The final pages provide a vast array of qualitative image completion examples at $64 \times 64$ resolution.

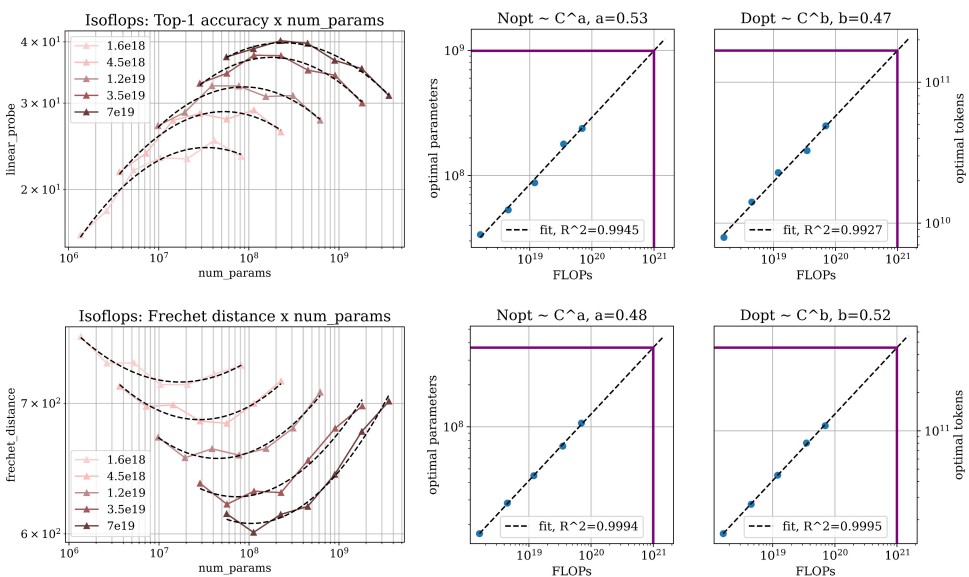

*Figure 6.* 32x32 IsoFLOPs, Top-1 accuracy (top row), Frechet Distance (bottom row) between generated samples and ground-truth images at native resolution.

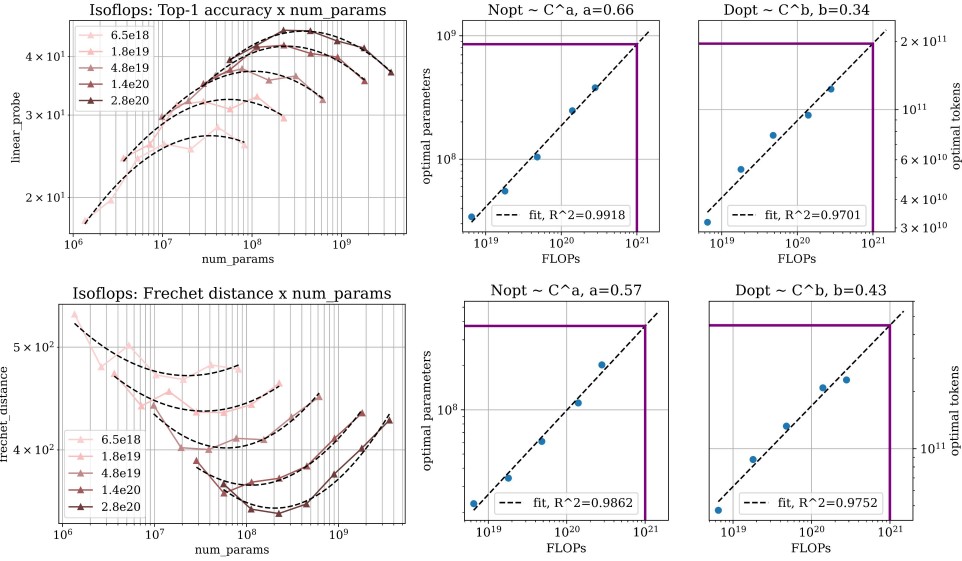

*Figure 7.* 64x64 IsoFLOPs, Top-1 accuracy (top row), Frechet Distance (bottom row) between generated samples and ground-truth images at native resolution.

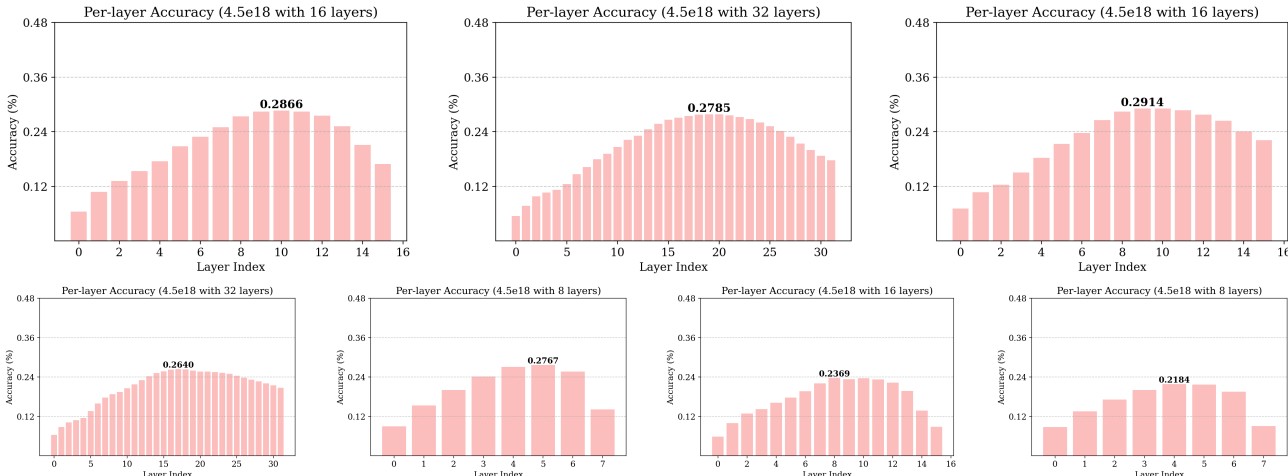

*Figure 8.* Per-layer linear probing accuracy with FLOPs budget of 4.5e18, trained on 32x32 image resolutions. A layout with 3 figures on the top row (base and IsoFLOP variant 0-1) and 4 figures on the bottom row (IsoFLOP variant 2-5).

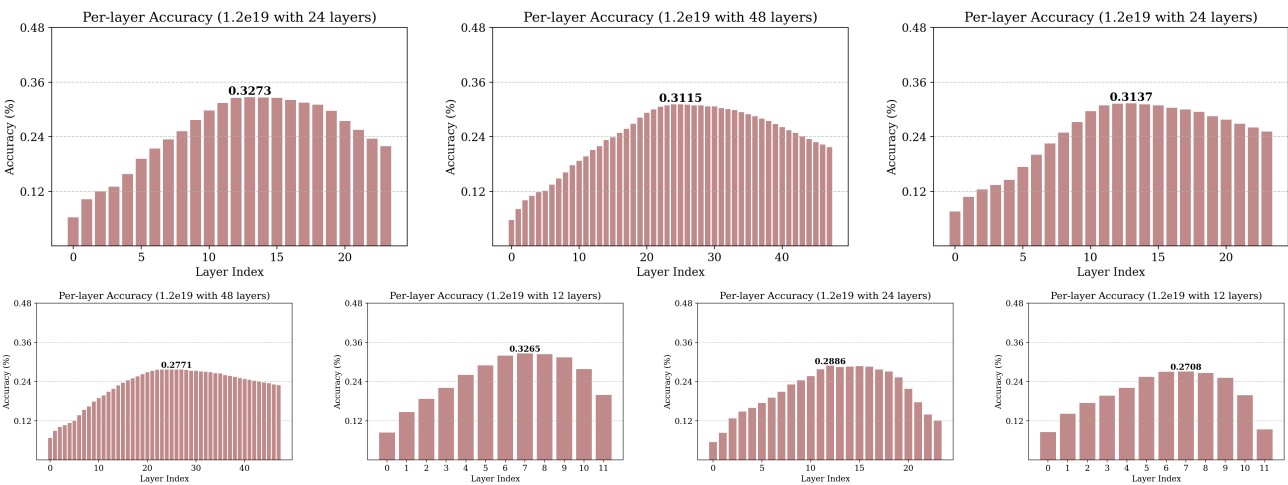

*Figure 9.* Per-layer linear probing accuracy with FLOPs budget of 1.2e19, trained on 32x32 image resolutions. A layout with 3 figures on the top row (base and IsoFLOP variant 0-1) and 4 figures on the bottom row (IsoFLOP variant 2-5).

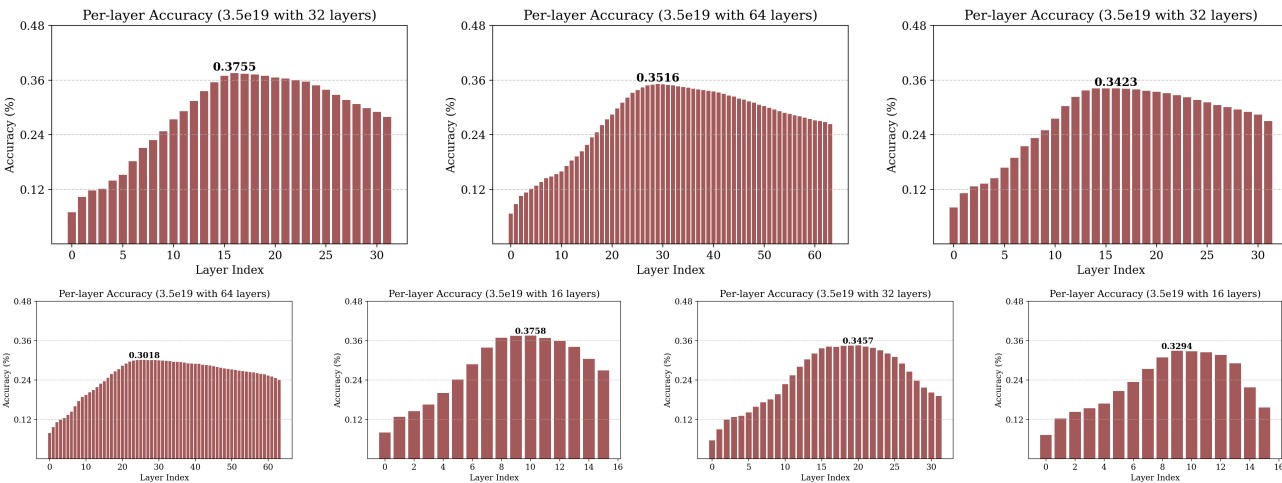

Figure 10. Per-layer linear probing accuracy with FLOPs budget of 3.5e19, trained on 32x32 image resolutions. A layout with 3 figures on the top row (base and IsoFLOP variant 0-1) and 4 figures on the bottom row (IsoFlOP variant 2-5).

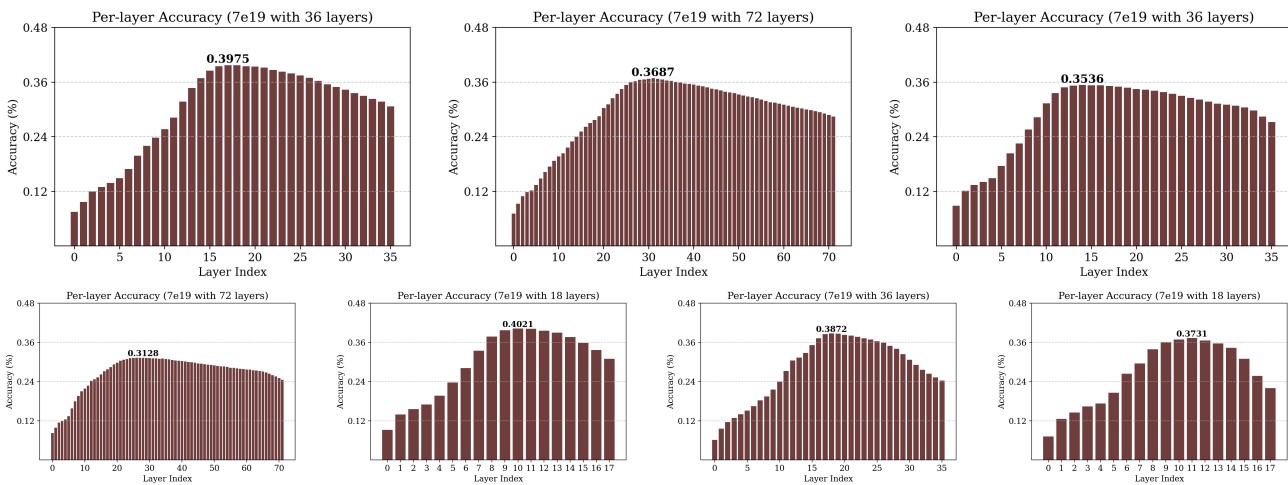

Figure 11. Per-layer linear probing accuracy with FLOPs budget of 7e19, trained on 32x32 image resolutions. A layout with 3 figures on the top row (base and IsoFLOP variant 0-1) and 4 figures on the bottom row (IsoFLOP variant 2-5).

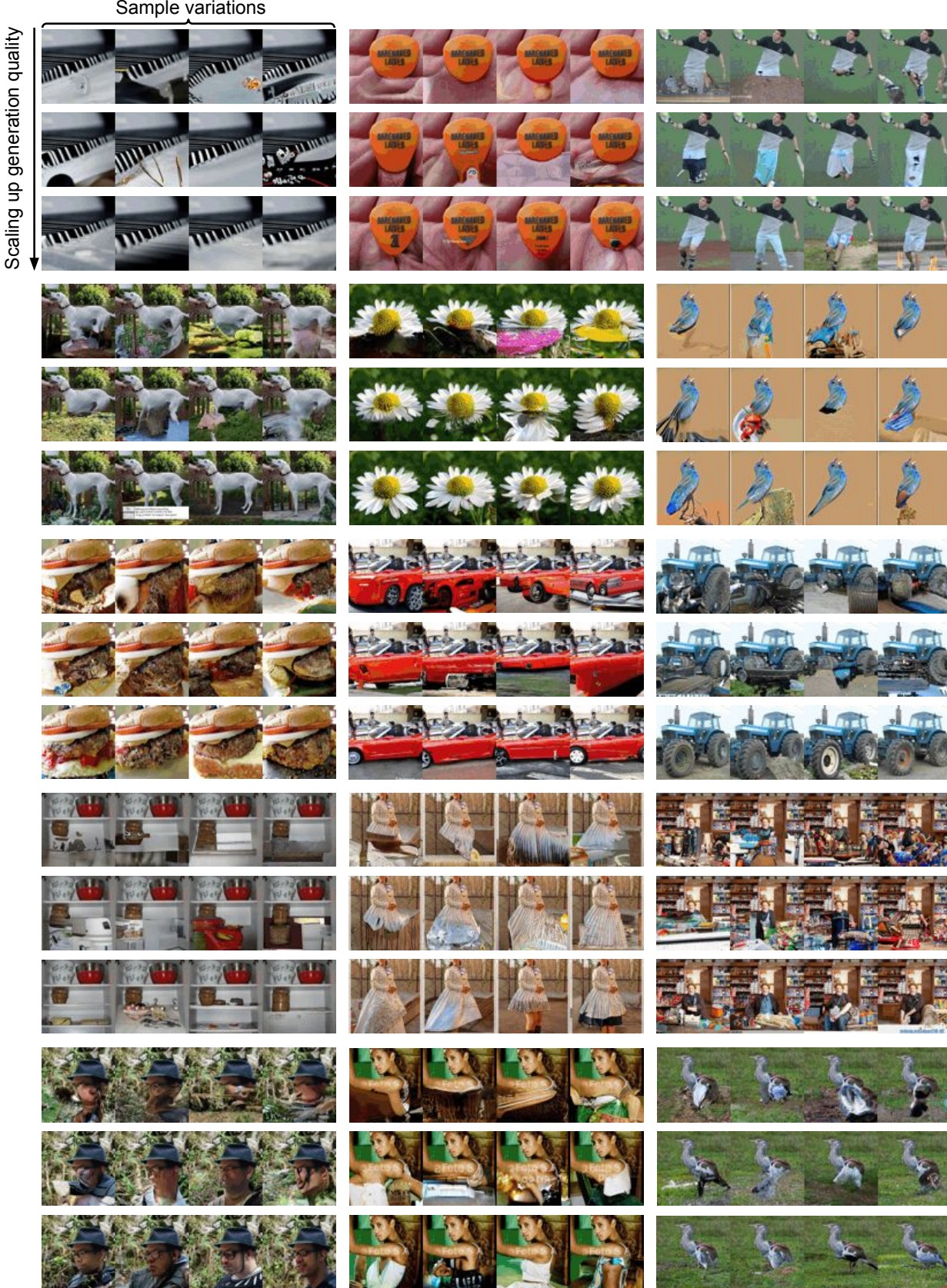

*Figure 12.* Qualitative examples at 64 × 64. The unmasked top half image is provided as initialization and we autoregressively predict the bottom half image one pixel at a time. Zoom in for a better view.

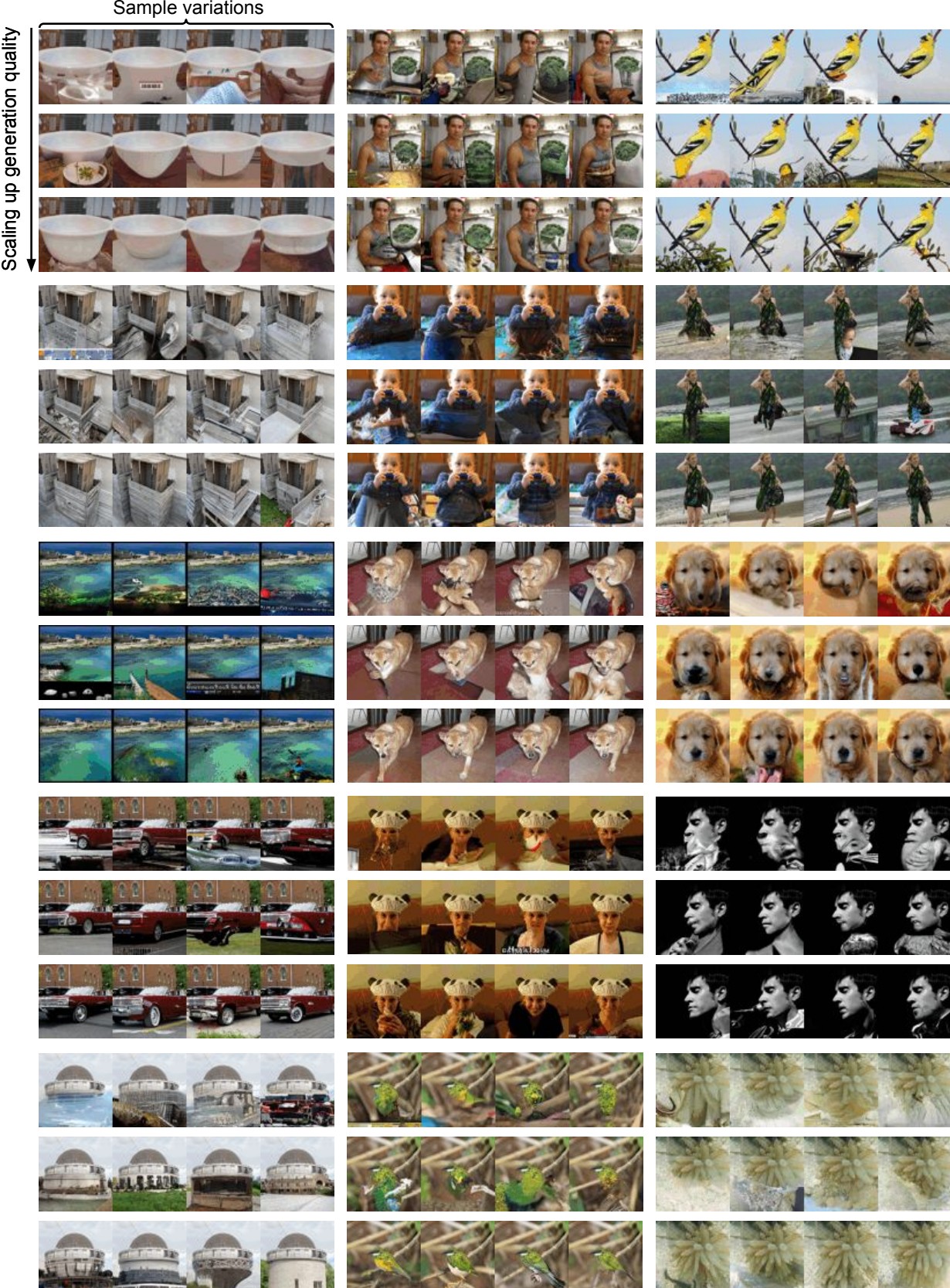

*Figure 13.* Qualitative examples at 64 × 64. The unmasked top half image is provided as initialization and we autoregressively predict the bottom half image one pixel at a time. Zoom in for a better view.

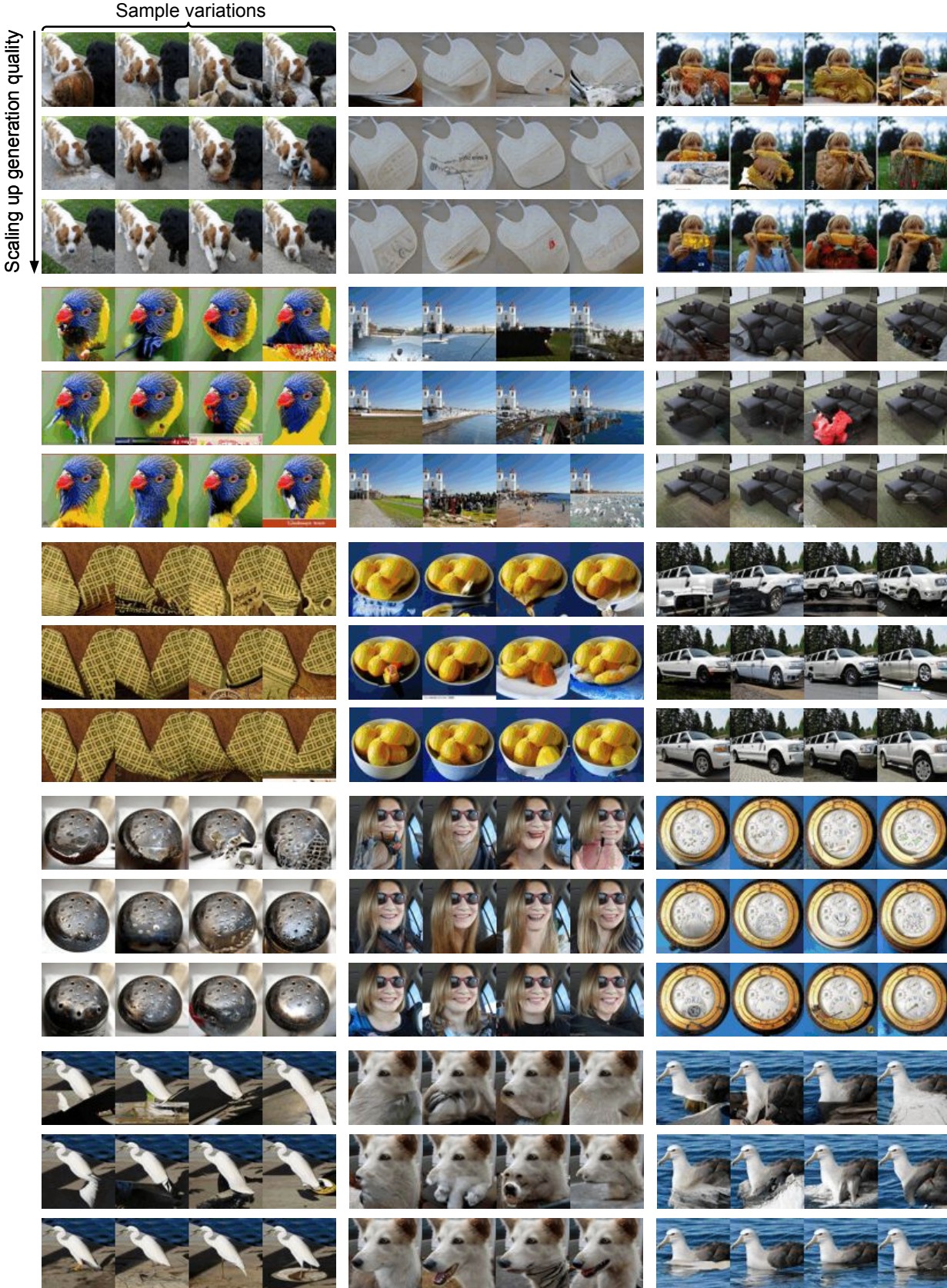

*Figure 14.* Qualitative examples at 64 × 64. The unmasked top half image is provided as initialization and we autoregressively predict the bottom half image one pixel at a time. Zoom in for a better view.

