# OpenReview forum: "Rethinking Generative Image Pretraining: How Far Are We From Scaling Up Next-Pixel Prediction?"
_ICML.cc/2026/Conference — ICML 2026 regular_

### Official Review · Reviewer_bjAW · 2026-02-23

**Soundness:** 3
**Presentation:** 3
**Significance:** 3
**Originality:** 3
**Overall Recommendation:** 5
**Confidence:** 4

**Summary:**

This manuscript investigates the scaling laws of autoregressive next-pixel prediction. First, the authors establish a Scaling Law framework for images, training a series of Transformers under compute budgets up to 7e19 FLOPs. Based on this, the authors find that the optimal scaling strategy is task-dependent. Under the same resolution, the optimal data size for image generation tasks needs to grow 3 to 5 times faster than for classification tasks. The authors also find that the optimal scaling strategy is resolution-dependent. As resolution increases, the optimal strategy requires model size to grow much faster than data size, showing that higher-resolution images provide richer learning signals. Finally, the authors provide forward-looking predictions and argue for their feasibility. By extrapolating the scaling curves, the authors predict that if compute grows by 10,000 times, pixel-by-pixel modeling could reach around 80% classification accuracy on ImageNet.

I think the authors' current arguments are careful and the experiments are solid. Moreover, the scaling properties of generative pretraining on raw pixels have indeed been undervalued. These make this research meaningful. However, I think the current paper has two major weaknesses. First, as the authors themselves admit, extrapolating from 16x16, 32x32, and 64x64 to 256x256 carries too much uncertainty. The prediction target is 10,000 times beyond the current compute range, which is really hard to find convincing. Second, the authors use completion-based Frechet Distance, meaning given one half of an image to complete the other half. This is too far from mainstream unconditional generation, making it hard to directly apply the conclusions to the image generation field. Despite these two issues, I think given the current lack of similar work and the guidance this work offers for the future, I tentatively give a recommendation of weak accept.

**Compliance With Llm Reviewing Policy:**

Affirmed.

**Final Justification:**

The authors' rebuttal successfully addressed my primary concerns. Thus, I raise my score.

**Key Questions For Authors:**

The questions I want to raise have already been listed in the weaknesses section and will not be repeated here.

**Limitations:**

yes

**Strengths And Weaknesses:**

Strengths:

1. The research problem is very valuable. Compared to image tokenizer plus autoregressive approaches, the scaling properties of end-to-end pixel-by-pixel modeling have indeed been seriously undervalued.

2. The experimental design is solid, and the IsoFlops Profile method used is a well-known approach.

3. The experimental results provide useful guidance for future work and benefit the community working on end-to-end pixel-by-pixel modeling.

Weaknesses:

Let me first point out some issues with the details of the manuscript, and then give a high-level assessment.

1. The Abstract right away describes next-pixel prediction as "a simple, end-to-end yet under-explored framework for unified vision models." I think "unified" may not be wrong. But the current manuscript only runs experiments on image classification and image completion generation, and at fairly low resolutions. This is far from the tasks we usually think of for unified vision models, such as detection, segmentation, and multimodal understanding. This claim is not well supported within this manuscript.

2. In the Abstract and Introduction, you repeatedly say you are measuring generation quality. But what you are actually doing is a completion task. I find it hard to agree with this framing. Or I think you should directly say that what you are doing is a generation-based completion task.

3. You say that "representing each image as pixel sequence enforces the least inductive bias on the structure of images." But at this point you have already introduced a prior assumption about image structure. Why does the ordering of pixels have to be raster-based rather than neighbor-based [1]? I think this expression is not quite right.

4. In Section 3.4, your claim of "infinite data limit" is too optimistic. X-300M has 300M images, and under high compute budgets it is quite possible for the data to be seen more than once.

5. Table 2 has several blank entries, and I am not sure what the blanks mean. Does it mean the values are kept the same? If so, what is the baseline they are kept the same relative to?

6. In Figure 2 you say you use "2048 reference images at 32x32 and corresponding 8192 generated images." But in Figure 4 this changes to "between 10,000 reference images and 10,000 generated images." Why are the standards different? Can the results from different standards both be generalized? I would like you to explain this.

The above covers some detail-level issues I see in the manuscript. Below are some high-level points I think need to be addressed.

7. The highest compute used in the current experiments is only 7e19 FLOPs, yet the predictions go up to 10,000 times that level. The conclusions are hard to find convincing. Using 5x compute as support for such a large extrapolation is also not very persuasive.

8. The manuscript uses completion-based Frechet Distance to measure generation quality. This is fundamentally different from unconditional generation metrics such as FID and IS, and the tasks are also different. This makes it hard to directly compare the method with approaches like LDM and DiT.

9. The description of X-300M is far too brief. What is the data source? What is the distribution of image categories? What image preprocessing steps were used? Since the quality and variety of the dataset directly affect the core conclusions, the fact that the dataset is not public greatly limits the impact of the contributions.

[1] Neighboring Autoregressive Modeling for Efficient Visual Generation, ICCV25

---

> ### Author Rebuttal · Authors · 2026-03-30
>
> We thank Reviewer bjAW for their thoughtful evaluation and for recognizing the value in scaling properties of end-to-end, raw pixel modeling. We are encouraged that the reviewer found our experimental design rigorous and our findings useful for the community. We also appreciate your careful read of our figures and tables; we will ensure the revised paper clarifies the blank entries in Table 2 and the different evaluation sample sizes in Figures 2 and 4. We appreciate reviewer constructive critiques regarding our compute extrapolations, terminology for generation and unified models, and dataset transparency, which we address below.
>
> ### [eYGX/1cxP/RhJM/bjAW] **Extrapolation to $1e24$ FLOPs** ###
> (See general response in eYGX (R1))
>
> ### [1cxP/bjAW] **Choice of Dataset / X-300M Dataset** ###
> (See general response in 1cxP (R2))
>
> ### [bjAW] **Evaluation Framework: Terminology, Metrics, and LDM/DiT Comparisons** ###
> We address the terminology, our chosen metrics, and the baseline comparisons below:
> - Terminology (Generation vs. Completion): Although our protocol uses conditional completion, we retain "generation" in the Abstract because autoregressive next-pixel prediction is fundamentally generative; the top-half completion acts as a deterministic "prompt". We will update the paper to explicitly state that generation quality is measured via completion to avoid confusion.
> - The Evaluation Metric (Completion FD vs. Standard FID):  We acknowledge that half-image completion differs from the unconditional generation typically used in mainstream evaluation. However, in the context of autoregressive modeling, unconditional generation is a special case of next-pixel prediction with empty context.
> We specifically chose a completion-based metric on a **fixed validation set** because the mainstream practice of calculating unconditional FID directly against the training set is unsuitable for establishing scaling laws. Specifically, it prevents the study of different training data mixtures during scaling, as the evaluation target constantly shifts with the data recipe. By providing a top-half spatial context per example from a held-out validation set, we establish a stable **semantic anchor**. This isolates the model's capacity to learn the visual distribution in a controlled environment, ensuring our measured trajectories reflect pure generative capability rather than shifting dataset statistics.
> - Comparisons against LDMs/DiTs: We recognize the asymmetry in directly comparing our unsupervised, unidirectional completion against bidirectional, latent-space diffusion models. To bridge this gap, one could evaluate both paradigms using class-conditioned generation. However, introducing class conditioning fundamentally breaks the core motivation of our framework: purely label-free next-pixel prediction. It is precisely this lack of reliance on class labels that allows our method to natively leverage internet-scale, open-world image distributions. Establishing standardized empirical scaling laws across these fundamentally different paradigms (unsupervised AR vs. conditioned Diffusion) is a massive undertaking, and we leave the formulation of comparative scaling laws for LDMs/DiTs as an important direction for future work.
> ### [bjAW] **Unified Vision Model Claim** ###
> We acknowledge our current evaluations on low-resolution classification and completion do not yet cover complex tasks like detection or segmentation.
> However, we retain the term *unified* as we have demonstrated discriminative and generative tasks for the framework. Our IsoFlops profiling is a foundational step toward this goal; detection and segmentation only become meaningful at resolutions of $224\times224$ or higher. Due to FLOP budget constraints limiting our analysis to $64\times64$ and below, benchmarking these specific tasks was not yet computationally practical.
> ### [bjAW] **Inductive Bias and Pixel Ordering** ###
> We agree that claiming 1D sequences enforce the "least inductive bias" was overbroad, as raster-scan ordering introduces a structural prior by separating vertical neighbors. We will revise our claim to "minimal architectural priors compared to patch/latent compression" and explicitly discuss [1] (Neighboring Autoregressive Modeling) as a vital future direction for reducing these sequential biases.
> ### [bjAW] **Infinite Data Assumptions** ###
> We agree that describing our empirical setup with X-300M dataset as operating in an "infinite data limit" is technically imprecise, and we will clarify this. This terminology was intended to characterize the capacity of the purely self-supervised, label-free framework, which is fundamentally unconstrained by the human-annotation bottlenecks that limit supervised methods. Because our next-pixel prediction approach can natively ingest the limitless supply of raw internet images (e.g., DataComp-1B), the assumption of an effectively infinite data distribution remains valid for the framework's scaling trajectory.

---

> > ### Author Rebuttal · Reviewer_bjAW · 2026-04-03
> >
> > Hello authors, thank you for taking the time to address my concerns.
> >
> > Your explanation regarding the choice of the evaluation metric (Completion FD vs. Standard FID) makes sense in the specific context of establishing scaling laws, where a stable evaluation target is necessary. It is also good to see that the overbroad claims regarding inductive bias and the "infinite data" limit will be refined in the revision.
> >
> > Please ensure that the clarifications provided here, as well as the important details concerning the dataset and extrapolation addressed in the general responses, are explicitly included in the final manuscript.
> >
> > Thank you for your hard work. I will raise my score.

---

### Official Review · Reviewer_RhJM · 2026-03-10

**Soundness:** 3
**Presentation:** 3
**Significance:** 3
**Originality:** 3
**Overall Recommendation:** 4
**Confidence:** 3

**Summary:**

This paper revisits autoregressive next-pixel prediction as a unified framework for vision modeling and investigates its scaling behavior with compute, model size, and dataset size. While recent generative vision models typically rely on tokenized representations for efficiency, the authors explore whether raw pixel autoregressive modeling can become competitive if scaled sufficiently.

**Compliance With Llm Reviewing Policy:**

Affirmed.

**Final Justification:**

The authors have satisfactorily addressed my concerns, and I will maintain my positive score.

**Key Questions For Authors:**

1. **Practical feasibility and scaling extrapolation.** The paper argues that the main bottleneck for pixel-level autoregressive modeling is compute rather than data, and extrapolates the observed scaling trends to suggest that such models may become competitive with significantly larger compute budgets. However, the current experiments are conducted within a relatively limited compute and resolution regime. Could the authors elaborate on the robustness of these extrapolations? In particular, how sensitive are the projected scaling trends to factors such as architectural choices, training dynamics, or resolution increases that may alter the token statistics and optimization landscape?

2. **Pixel-space vs. latent-space generative modeling.**  While the paper focuses on understanding the scaling behavior of pixel-level autoregressive models, it would be helpful to better contextualize these results relative to latent-space generative approaches that dominate modern image generation. Under sufficiently large compute budgets, how does naive pixel-level modeling compare with latent-based methods in terms of generation quality and efficiency? If the performance remains inferior, what would be the key advantages of pixel-space modeling that justify pursuing this direction?

**Limitations:**

**No.** While the paper provides an extensive empirical study, it does not explicitly discuss several important limitations. In particular, the experiments are restricted to relatively low image resolutions, and it remains unclear whether the observed scaling behaviors would generalize to more realistic high-resolution settings (e.g., 256×256 or 512×512). A discussion of how resolution scaling might affect the conclusions, as well as potential computational and practical limitations of pixel-level modeling at high resolutions, would strengthen the paper.

**Strengths And Weaknesses:**

Strengths：

1. The paper studies the scaling behavior of pixel-level autoregressive models, addressing an important and underexplored question in generative vision research.

2. The experiments are extensive and carefully designed, providing a thorough empirical analysis across model size, dataset size, compute budgets, and multiple evaluation metrics.

Weaknesses：

1. Although the experimental study is extensive, it mainly focuses on low-resolution images (e.g., 32×32), leaving it unclear whether the same scaling trends would hold at more realistic resolutions such as 256×256 or 512×512.

---

> ### Author Rebuttal · Authors · 2026-03-30
>
> We thank Reviewer RhJM for their constructive feedback and for recognizing that our study addresses an important, underexplored question in generative vision research. We are encouraged by your positive assessment of our extensive and carefully designed empirical analysis across various compute budgets, model scales, and evaluation metrics. We acknowledge your questions regarding the robustness of our scaling extrapolations to high resolutions and the conceptual comparison to latent-space models, which we address in detail below.
>
> ### [eYGX/1cxP/RhJM/bjAW] **Extrapolation to $1e24$ FLOPs** ###
> (See general response in eYGX (R1))
>
> ### [eYGX/1cxP/RhJM] **High-Resolution Scaling and the "Compute-Bound" Conclusion** ###
> (See general response in eYGX (R1))
>
> ### [RhJM] **Robustness of Scaling Trends to Architecture and Training Dynamics** ###
>
> Thanks for pointing out that scaling to practical resolutions introduces new variables in training dynamics and architectural design. However, drawing upon established scaling law literature, we anticipate our fundamental scaling trends (the power-law exponents) to be highly robust to these factors:
> - Training Dynamics and Optimization: While the optimization landscape inherently shifts at larger scales, the foundational scaling laws of autoregressive modeling are remarkably resilient. Research in language modeling (e.g., Kaplan et al., 2020) has consistently demonstrated that while suboptimal hyperparameters might shift the absolute performance curve upward (worsening the intercept), the overarching **scaling exponent remains stable**.
> - Architectural Choices: Similarly, the literature shows that while specific architectural configurations (such as depth-to-width ratios or attention head routing) can improve compute-efficiency ceilings, they rarely alter the fundamental power-law trajectory of the data modality itself.
> Therefore, we strictly constrain our empirical claims to the resolutions evaluated in this study. However, we hypothesize that the core “compute-bound” nature of raw pixel modeling will persist at $256 \times 256+$ resolutions. Validating whether the fundamental scaling exponent remains robust is a critical future direction for this research trajectory.
>
> ### [RhJM] **Pixel-Space vs. Latent-Space Generative Modeling** ###
> To answer directly: under current compute budgets, latent-based methods (like LDMs) are indeed significantly more efficient and generally achieve higher generation quality. By utilizing pre-trained VAEs, they compress spatial redundancy, drastically reduce sequence lengths, and bypass the need to model high-frequency pixel noise.
>
> However, while latent models are highly optimized for current constraints, our primary objective is to study the scaling laws of raw pixel prediction under sufficiently large compute regimes. The key advantages of mapping this foundational, tokenizer-free trajectory include:
> - Eliminating Information Bottlenecks: Pre-trained autoencoders and discrete tokenizers are inherently lossy. They introduce permanent information bottlenecks (e.g., permanently discarding fine-grained, high-frequency details) and inject domain biases from their own pre-training distributions. Raw pixel modeling operates directly on the fundamental visual signal, guaranteeing a lossless representation that forces the model to natively learn both global semantics and microscopic details end-to-end.
> - Providing a Pure Scientific Baseline: While structural priors (e.g., frozen latent spaces) are highly effective for accelerating training, they act as confounding variables when studying scaling laws. Pixel-level autoregression makes absolute minimal architectural assumptions. By operating strictly on the raw data distribution, it provides a domain-agnostic baseline to study how visual modeling scales fundamentally with compute, independent of any specific compression scheme.
> Therefore, while our current empirical study is bounded by computational limits, we view mapping these raw-pixel scaling laws not as a replacement for LDMs, but as a necessary scientific baseline for understanding tokenizer-free visual intelligence. Ultimately, your question highlights a critical open problem: conducting a concrete scaling comparison between pixel-level and latent-space paradigms is a valuable direction for future research.

---

> > ### Author Rebuttal · Reviewer_RhJM · 2026-04-03
> >
> > Thank you for the thorough and well-prepared rebuttal. The authors have satisfactorily addressed my concerns, and I will maintain my positive score.

---

### Official Review · Reviewer_1cxP · 2026-03-12

**Soundness:** 3
**Presentation:** 4
**Significance:** 3
**Originality:** 3
**Overall Recommendation:** 5
**Confidence:** 4

**Summary:**

This paper investigates the scaling properties of autoregressive next-pixel prediction as a general training framework for vision models. The key issue addressed by this paper is understanding how far we are from making raw pixel-level modeling practical at scale. The work aims to study the concept of compute-optimal scaling for next-pixel prediction across both recognition and generation tasks. Using IsoFlops profiles, the authors train Transformers on images across compute budgets up to 3.5e20 FLOPs and evaluate three metrics: next-pixel prediction loss, ImageNet linear probing accuracy, and completion-based Fréchet Distance.

The main findings are: (1) pixel prediction requires 10-20× higher token-to-parameter ratios than language tokens; (2) optimal scaling strategies are task-dependent, with generation requiring 3-5× faster data growth than classification; (3) at higher resolutions, model size should grow faster than data size; (4) the bottleneck is compute rather than data availability, with projections suggesting feasibility within five years.

**Compliance With Llm Reviewing Policy:**

Affirmed.

**Final Justification:**

To my knowledge, this work is the first systematic scaling law study comparing recognition and generation tasks under the same next-pixel prediction framework. It provides novel hindsight such as optimal scaling strategies diverge across tasks (recognition vs. generation) at fixed resolution, which has implications for unified vision model design, pixel prediction requires 10-20× higher token-to-parameter ratios than language tokens. During rebuttal, authors addressed my main concern regarding the extrapolation to 1e24 Flops.

**Key Questions For Authors:**

How sensitive are the scaling exponents to the choice of dataset? Would training on ImageNet alone yield different optimal strategies?
Have you observed any signs of scaling law breakdown at the higher end of your compute range?

**Limitations:**

yes

**Strengths And Weaknesses:**

Strength:
- The paper is very well-written with clear narrative flow and well-designed figures (especially Figure 1 and Figure 4) that effectively communicate key findings.
- To my knowledge, this work is the first systematic scaling law study comparing recognition and generation tasks under the same next-pixel prediction framework. It provides novel hindsight such as optimal scaling strategies diverge across tasks (recognition vs. generation) at fixed resolution, which has implications for unified vision model design, pixel prediction requires 10-20× higher token-to-parameter ratios than language tokens,
- The compute-bound finding (rather than data-bound) provides a clear roadmap for future scaling efforts.


Weaknesses:
- Experiments are primarily conducted at 32×32 resolution, with limited ablations at 16×16 and 64×64. Extrapolating to practical resolutions (224×224+) involves significant uncertainty as the observed trends may not hold.
- Model sizes cap at 449M parameters, whereas modern vision transformers scale to 22B+. Scaling dynamics could change qualitatively at larger scales.
- The X-300M dataset is not described beyond its size. Dataset composition, diversity, and how it compares to standard benchmarks remain unclear.
- Projecting to 10^24 FLOPs (10,000× beyond tested range) is highly speculative. Power laws could potentially break down at such extrapolations.

---

> ### Author Rebuttal · Authors · 2026-03-30
>
> We are grateful to Reviewer 1cxP for their thorough evaluation and for highlighting the significance of comparing recognition and generation within a single scaling law framework. It is highly encouraging that the clarity of our narrative, our figure design, and the roadmap established by our compute-bound findings resonated with your assessment. At the same time, we acknowledge your valid concerns regarding dataset transparency, potential scaling bottlenecks, and extrapolation risks, which we address comprehensively below.
>
> ### [eYGX/1cxP/RhJM/bjAW] **Extrapolation to $1e24$ FLOPs** ###
> (See general response in  eYGX (R1))
>
> ### [eYGX/1cxP/RhJM] **High-Resolution Scaling and the "Compute-Bound" Conclusion** ###
> (See general response in eYGX (R1))
> ### [1cxP] **Model Size Cap at 449M Parameters vs. 22B+ Models** ###
> We agree that our maximum model capacity of 449M parameters is smaller than modern 22B+ vision transformers, and we acknowledge that scaling dynamics can shift qualitatively at massive scales. However, it is important to contextualize the architectural differences. Modern 22B+ models are typically encoder-only, trained using patch-level tokenizations with classification objectives (e.g., ViT-22B uses only 256 visual tokens per image). In contrast, our framework performs dense, next-pixel auto-regressive prediction completely **unsupervised** (does not require a dataset with human labels). Executing our required IsoFlops parallel training runs at multi-billion parameter scales was computationally prohibitive given our FLOPs budget ($7 \times 10^{19}$ at $32 \times 32$ and $2.8 \times 10^{20}$ at $64 \times 64$). Our goal was to map the foundational scaling trajectory; we will explicitly highlight this 449M parameter ceiling in the limitations section, noting that validating these power laws against the emergent dynamics of massive-scale models remains a critical future milestone.
> ### [1cxP/bjAW] **Choice of Dataset / X-300M Dataset** ###
> We thank Reviewers 1cxP (R2) and bjAW (R4) for highlighting the need for greater transparency regarding the X-300M dataset. We agree that dataset composition and diversity are critical factors that directly influence scaling laws.
> The brief description in our initial submission was a conservative precaution to strictly adhere to the double-blind review policy, as explicitly naming or citing the literature for this specific dataset could compromise anonymity. However, we are happy to provide the core statistics and distribution details here.
> X-300M is a highly diverse, large-scale dataset consisting of approximately 300 million web-sourced images.
> - Distribution vs. Standard Benchmarks: Compared to standard benchmarks like ImageNet (which enforces a strictly uniform distribution across 1,000 largely object-centric classes), X-300M features a heavily long-tailed distribution. This naturally skewed distribution much more accurately reflects open-world visual frequencies, encompassing a vast variety of complex scenes, natural landscapes, text-heavy images, and diverse objects. We utilized this specific mixture precisely to ensure our scaling laws generalized to "in-the-wild" visual modeling, rather than overfitting to an ImageNet training set.
> ### [1cxP] **Training on ImageNet Alone & Scaling Law Breakdowns** ###
> We thank Reviewer 1cxP (R2) for this excellent question. In fact, we initially began our preliminary scaling studies by training exclusively on ImageNet.
> However, we found the model overfit extremely early when using the simple data augmentations adopted in early pixel-prediction works (such as iGPT). While contemporary training methods often leverage massive data augmentations to prevent overfitting, doing so introduces a major confounding variable into the scaling formulation. Our primary objective was to study pure architectural scaling decoupled from augmentation engineering. This necessity to isolate pure scaling behavior was our central motivation for transitioning to the inherently diverse, internet-scale X-300M dataset.
> Regarding potential scaling law breakdowns: within our explicitly tested compute budget, we did not observe any statistically significant deviation from the power-law trajectories. However, we completely agree that as models scale further toward massive parameter counts, they will potentially encounter structural breakdowns.

---

> > ### Author Rebuttal · Reviewer_1cxP · 2026-04-01
> >
> > Authors addressed my main concern regarding the extrapolation to 1e24 Flops and resolutions and will revise the paper accordingly.

---

### Official Review · Reviewer_eYGX · 2026-03-12

**Soundness:** 2
**Presentation:** 4
**Significance:** 3
**Originality:** 3
**Overall Recommendation:** 4
**Confidence:** 4

**Summary:**

The paper is mainly asking a simple question: how far can we scale raw next-pixel prediction for images? In other words, it studies the possibility and feasibility of learning directly from pixels, without tokenizers or patch-based representations.

The challenge is that next-pixel prediction is much harder than language modeling: individual pixels carry very little semantic meaning, image pixels have complex spatial dependencies, and the sequence length grows rapidly as image resolution increases.

To study this, the paper uses a fixed FLOPs budget and scaling-law analysis to estimate the compute-optimal amount of data and model size. It evaluates scaling behavior with three targets: next-pixel prediction loss, ImageNet classification accuracy, and generation quality measured by Frechet Distance.

The main claims are:

1. the optimal scaling strategy depends on the target task;
2. as image resolution increases, model size needs to grow much faster than data size;
3. the main bottleneck is compute, not the amount of training data.

**Compliance With Llm Reviewing Policy:**

Affirmed.

**Final Justification:**

The authors' rebuttal has adequately addressed my main concerns. I am inclined to accept this paper.

**Key Questions For Authors:**

1. Why should the long-range extrapolation to much larger models and compute budgets be considered reliable table 3 and 4?

2. Could the authors clarify whether the current IsoFlops study, with relatively few runs and model scales, is sufficient to support the claimed scaling trends, especially compared with prior large-scale studies such as Chinchilla that used around 400 runs?

3. How strongly is the “compute-bound rather than data-bound” conclusion supported beyond the low-resolution regime studied here?

4. Figure 4 is central to the claim that higher resolutions favor faster model scaling than data scaling. Have the authors considered a resolution-transfer setting, where a model is first trained at 32x32 and then directly evaluated, or lightly continued training, at 64x64? If such a progressive-resolution curve lies above the current from-scratch high-resolution curve, it would suggest that the projected compute requirements in Figure 4 may be overstated.

**Limitations:**

Yes

**Strengths And Weaknesses:**

## Soundness:

The motivation is well established and grounded in both prior literature and domain knowledge. The paper’s main claims are also supported to a fair extent by a fairly comprehensive set of results. For example, the claim that the optimal scaling strategy is task-dependent is supported by the experimental results, especially those summarized in Figure 1. I also think it is important to view this paper correctly: it is not primarily a new scaling-law paper, but rather a study of image understanding and generation through the lens of scaling laws. That said, the final forward-looking claim, namely that raw next-pixel prediction may become feasible within the next five years, is much less convincing because it relies heavily on extrapolation, and the extrapolation extends too far beyond the directly tested regime.
The experimental design generally follows established scaling-law methodology from NLP, including the use of IsoFlops analysis, and the model family is reasonably comparable to iGPT-style architectures. The study mainly focuses on images from 16x16 to 64x64, which is a sensible range given the computational cost of pixel-level autoregressive modeling. Overall, I would describe the experimental design as fairly rigorous. My main concern is that the core IsoFlops study is still based on a relatively small number of training runs and model scales, which limits how strongly one can trust the longer-range projections.



## Presentation:

The writing of this paper is very good and content is easy to follow. The structure of this paper is well orginzied. One minor concern is Figure 4 should put in front of Figure 3 and Table 4?


## Significance:

The paper studies an important and relevant problem. This question matters because most current generative vision systems rely on tokenizers, patchification, or multi-stage pipelines, so understanding the true scaling behavior of direct pixel modeling is valuable. The main findings are also meaningful, since they provide concrete guidance on how scaling differs across tasks and resolutions, and suggest that compute, rather than data, is the main bottleneck. Although the immediate practical impact is limited by the low-resolution setting and the heavy reliance on extrapolation, the paper can still benefit future research by informing how model size, data, and compute should be allocated in future raw-pixel or unified generative vision models



## Novelty:

The paper has moderate novelty. Its main originality lies in the empirical findings rather than the method itself. In particular, the observations that the optimal scaling strategy differs across next-pixel loss, classification, and generation, that higher resolutions favor faster model scaling than data scaling, and that raw next-pixel prediction is more compute-bound than data-bound are the more novel parts. In contrast, the method uses a standard autoregressive next-pixel prediction setup, standard Transformer architectures similar to iGPT, and established IsoFlops analysis.

---

> ### Author Rebuttal · Authors · 2026-03-30
>
> We thank Reviewer eYGX (R1) for the review and for explicitly recognizing our work as an exploration of image understanding and generation through the lens of scaling laws. We appreciate your positive assessment of our experimental design, excellent presentation, and the broader significance of our compute-bound findings. We also agree with your presentation feedback and will swap the placement of Figure 4 to appear before Figure 3 and Table 4 to improve the narrative flow. Your critiques regarding our extrapolation boundaries and the potential for resolution-transfer are insightful, and we address each question below.
>
> ### [eYGX] **Limited Training Runs and Scales** ###
> While our run count is lower than Chinchilla, our run density is proportionally rigorous for the compute regime studied. Specifically, Chinchilla used ~400 runs over three orders of magnitude. Our study uses 105 runs across a concentrated range ($4 \times 10^{17}$ to $2.8 \times 10^{20}$ FLOPs) and three resolutions, evaluating 5 compute ladders per resolution with 7 IsoFlops variants each (see Figure 1/2/3 in the supp).
> These variants reliably identify compute-optimal minima. We acknowledge risks in $10,000\times$ extrapolation; our revision will detail our run density relative to NLP studies and frame the $10^{24}$ FLOP projection as an estimate based on current empirical trajectories.
> ### [eYGX/1cxP/RhJM/bjAW] **Extrapolation to $1e24$ FLOPs** ###
> We completely agree with the reviewers that extrapolating four orders of magnitude ($10,000\times$) beyond our empirically tested FLOPs is inherently speculative. As noted, scaling from a 449M parameter ceiling to 22B+ models will likely introduce entirely new structural bottlenecks, architectural sensitivities, and optimization challenges.
> However, these projections are intended as data-driven theoretical estimates, not definitive guarantees. This methodology is standard practice for evaluating autoregressive scaling limits; notably, the [GPT-4 Technical Report](https://arxiv.org/abs/2303.08774) successfully utilized identical power-law fitting to predict validation loss across this exact same gap (four orders of magnitude).
> Our fundamental objective with this projection is to illustrate a broader theoretical takeaway: under established scaling laws, raw pixel modeling will remain constrained by computational capacity, not data availability, over a five-year horizon.
> We will revise the discussion surrounding Tables 3 and 4 to explicitly address these extrapolation risks and discuss potential scaling discontinuities at extreme parameter scales.
> ### [eYGX/1cxP/RhJM] **High-Resolution Scaling and the "Compute-Bound" Conclusion** ###
> We completely agree that extrapolating our scaling trends to practical resolutions (e.g., $224\times224+$) involves significant uncertainty. Because sequence lengths grow quadratically with resolution, models may encounter entirely new bottlenecks that cannot be confidently predicted using only low-resolution data.
> For this precise reason, the projections presented in our paper (e.g., Figure 4, Tables 3 & 4) are intentionally confined **strictly to the resolutions we empirically evaluated** ($16\times16$ to $64\times64$). We do not claim our specific empirical trajectories will seamlessly generalize to high-definition imagery.
> As suggested by R3, we will revise the "Limitations and Future Work" section to explicitly state our $\le 64\times64$ empirical boundary, discuss the computational challenges of scaling to practical resolutions, and frame the verification of the compute-bound paradigm at $224\times224+$ as a critical open question for future research.
>
> ### [eYGX] **Resolution growing** ###
> We agree that progressive resolution-transfer could improve token-to-parameter allocation, meaning our projections in Figure 4 likely represent an **upper bound** on required compute. However, evaluating this presents immediate technical and theoretical bottlenecks:
> - **Architectural constraints**: Our models use a 1D sequence index for positional embedding. Increasing resolution alters the underlying 2D spatial mapping, making direct fine-tuning sub-optimal without task specific modifications.
> - **Theoretical complexity**: Establishing **multi-stage scaling laws** requires jointly optimizing new variables, such as cross-resolution learning rate schedules and precise compute switch-points.
> Scaling laws for progressive sequence-length curricula are an open problem even in LLM research. In visual generation, preserving 2D spatial structures across 1D sequence expansions adds further complexity. We prioritize scaling current models straightforwardly to establish a baseline. While more sophisticated scaling methods exist, their design choices require further research and are left for future work.

---

> > ### Author Rebuttal · Reviewer_eYGX · 2026-04-03
> >
> > Thanks for the rebuttal,  my concerns have been solved and I will maintain the positive score.

---

### Decision · Program_Chairs · 2026-04-30

**Decision:**

Accept (regular)

**Comment:**

This paper investigates the scaling properties of autoregressive next-pixel prediction — a simple, end-to-end, tokenizer-free framework for unified vision models.

### Reviewer eYGX initially had the following concerns:
* final forward-looking claim (feasibility within five years) relies heavily on extrapolation that extends too far beyond the tested regime. Why should long-range extrapolation to much larger compute budgets be considered reliable?
* core IsoFlops study is based on relatively few training runs and model scales. Is the IsoFlps study (with relatively few runs) sufficient compared to prior large-scale studies like Chinchilla (~400 runs)?
* limited to images 16×16 to 64×64. How strongly is the "compute-bound rather than data-bound" conclusion supported beyond the low-resolution regime? Have authors considered resolution-transfer (train at 32×32, evaluate at 64×64)?

#### The rebuttal
* defended run density as proportionally rigorous within the tested compute regime (105 runs across 4 orders of magnitude, 5 compute ladders per resolution, 7 IsoFlops variants each)
* acknowledged risks in 10,000× extrapolation; will revise to frame projections as estimates
* agreed on extrapolation risks; will add an explicit Limitations section
* acknowledged resolution-transfer suggestion but cited architectural constraints and theoretical complexity

This response does not fully convince the AC, and agrees that a weak accept score might be sufficient.

### Reviewer 1cxP had some similar concerns:
* Experiments primarily at 32×32 resolution; extrapolation to 224×224+ involves significant uncertainty
* Model sizes cap at 449M parameters; scaling dynamics could change at larger scales
* Projecting to 10²⁴ FLOPs (10,000× beyond tested range) is highly speculative

#### The rebuttal
* acknowledged 449M ceiling; contextualized architectural differences (encoder-only vs. dense pixel AR prediction); committed to adding limitations section
* provided detailed X-300M dataset description
* convinced the reviewer of the main concern regarding the extrapolation to 1e24 Flops

### Reviewer RhJM shared similar concerns
* Practical feasibility and scaling extrapolation & Pixel-space vs. latent-space generative modeling.

#### The rebuttal
* anticipated fundamental scaling exponents (power-law) remain robust based on established scaling law literature, even if specific architectural choices shift the absolute curve.
* acknowledged latent-based methods (LDMs) are more efficient under current compute; defended pixel-level modeling value: (1) eliminates information bottlenecks from tokenizers, (2) provides a pure scientific baseline for tokenizer-free visual intelligence.

### Reviewer bjAW had some similar concerns and raised the score after the rebuttal.
Details are omitted here for brevity.